# From movement to cognitive maps: recurrent neural networks reveal how locomotor development shapes hippocampal spatial coding

**Marco P Abrate**[1], **Laurenz Muessig**[1], **Joshua P Bassett**[2], **Hui Min Tan**[3],
**Francesca Cacucci**[2], **Thomas J Wills**[1,†], **Caswell Barry**[1,†]

[1]Dept. of Cell and Developmental Biology; University College London
[2]Dept. of Neuroscience, Physiology, and Pharmacology; University College London
[3]Animal Concerns Research & Education Society of Singapore
[†]Equal contribution.

Correspondence: {marco.abrate, t.wills, caswell.barry}@ucl.ac.uk

## Abstract

The hippocampus contains neurons whose firing correlates with an animal's location and orientation in space. Collectively, these neurons are held to support a cognitive map of the environment, enabling the recall of and navigation to specific locations. Although recent studies have characterised the timelines of spatial neuron development, no unifying mechanistic model has yet been proposed. Moreover, the processes driving the emergence of spatial representations in the hippocampus remain unclear (Tan et al., 2017). Here, we combine computational analysis of postnatal locomotor development with a recurrent neural network (RNN) model of hippocampal function to demonstrate how changes in movement statistics – and the resulting sensory experiences – shape the formation of spatial tuning. First, we identify distinct developmental stages in rat locomotion during open-field exploration using published experimental data. Then, we train shallow RNNs to predict upcoming visual stimuli from concurrent visual and vestibular inputs, exposing them to trajectories that reflect progressively maturing locomotor patterns. Our findings reveal that these changing movement statistics drive the sequential emergence of spatially tuned units, mirroring the developmental timeline observed in rats. The models generate testable predictions about how spatial tuning properties mature – predictions we confirm through analysis of hippocampal recordings. Critically, we demonstrate that replicating the specific statistics of developmental locomotion – rather than merely accelerating sensory change – is essential for the emergence of an allocentric spatial representation. These results establish a mechanistic link between embodied sensorimotor experience and the ontogeny of hippocampal spatial neurons, with significant implications for neurodevelopmental research and predictive models of navigational brain circuits. *

## 1 Introduction

The hippocampal formation is a critical structure in the mammalian brain, necessary to support spatial memory and navigation. The spatially modulated neurons it contains, which provide a representation of self-location and orientation, are widely thought to be the basis of a cognitive map supporting flexible navigation through the environment (Tolman, 1948; O'keefe & Nadel, 1978). Perhaps best known of these, place cells are pyramidal neurons of hippocampal regions CA3 and CA1 which fire selectively when an animal occupies particular locations. These cells provide a sparse, allocentric (i.e., world-centred and independent of the observer's own position) mapping of the environment that persists even when individual environmental cues are altered or removed, suggesting they encode spatial position rather than individual sensory features (O'Keefe & Dostrovsky, 1971; O'Keefe & Conway, 1978). Complementing this positional code, head direction cells –

---

*Code available at: https://github.com/marcoabrate/movement2cogmaps

found primarily in the subicular complex and anterior thalamus – signal orientation, firing when the animal faces specific allocentric directions (Taube et al., 1990). Within this navigational network, additional spatial cell types contribute to the brain's representation of space. Boundary-responsive cells respond when the animal is at specific distances and directions from borders (Savelli et al., 2008; Solstad et al., 2008; Lever et al., 2009), while grid cells in the medial entorhinal cortex exhibit distinctive hexagonal firing patterns that systematically tile the environment (Hafting et al., 2005). Neurons with properties of multiple spatial coding categories have also been observed throughout the hippocampal formation, highlighting the complex and distributed nature of spatial processing.

The post-natal development of these neurons in rats follows a specific timeline that has been well characterised in recent studies (Wills et al., 2010; Langston et al., 2010; Tan et al., 2017). Head direction cells emerge earliest and can be detected in an immature form as early as postnatal days (P) 11 and 12 (Tan et al., 2015; Bjerknes et al., 2015), with adult-like cells emerging at P15 (Wills et al., 2010; Langston et al., 2010). Spatially-localised firing can be observed in place cells as early as P14 (Muessig et al., 2015), but this is less spatially specific and stable than that observed in adults. These characteristics then slowly mature over the next 3-4 weeks (Scott et al., 2011). Grid cells are the last spatially-modulated neurons to emerge in the rat, with stable and periodic grid firing first observable at P20 (Wills et al., 2010), and the transition to adult-like grid patterns occurs remarkably rapidly – over approximately 24 hours (Wills et al., 2012). The mechanisms driving the maturation of spatially tuned neurons in the hippocampus and the relative timing of their emergence remain unclear (Tan et al., 2017). In this study, we hypothesise that the sequential nature of sensory and motor experiences fundamentally influences the emergence and maturation of spatial representations in the hippocampus.

The postnatal development of locomotion in laboratory rats involves the maturation of both postural and locomotor skills. According to Altman & Sudarshan (1975), until P8 uncoordinated hindlimbs result in forelimb-driven pivoting movements. Between P8 and P14, quadrupedal walking begins to emerge, although hindlimbs still lag behind. From P14, this raised posture becomes dominant, with locomotor speeds and distances travelled continuing to increase throughout the third week of life (Gerrish & Alberts, 1996). The temporal alignment between locomotor and spatial neuron development is striking, suggesting a potential causal relationship. However, manipulations of the postnatal motor system that could test our hypothesis are difficult to implement and often require invasive procedures such as tenotomy, neurotomy, and nerve crushing (Lowrie & Vrbova, 1992). An alternative approach is computational modelling, which allows the relationship between behaviour, sensory sampling of the environment, and neuronal representations to be more easily manipulated.

Animals perceive their environment through sensory organs that provide inherently egocentric (i.e., self-centred) information – what is seen, heard, or felt from the animal's current position and orientation. The transformation of egocentric information into a coherent world-centred (allocentric) representation is crucial to support flexible navigation strategies such as taking shortcuts or planning routes to unseen locations (Tolman, 1948). The predictive learning framework positions the hippocampal formation as the system that compares incoming sensory stimuli with memory-derived predictions (Eichenbaum, 2004; Levy, 1989), and it is supported by observations that hippocampal neurons encode not just current location but trajectories of possible future positions (Johnson & Redish, 2007; Kay et al., 2020; Ujfalussy & Orbán, 2022). The integration of egocentric sensory information within this framework represents a fundamental computational challenge in spatial cognition: how to construct a stable internal model of space from a sequence of observations (Stachenfeld et al., 2017; Buzsáki & Tingley, 2018). Among sensory modalities, vision appears particularly important for this process in many mammals, providing rich spatial information that extends beyond immediate physical contact. Further, the influence of visual cues on spatial representations is evident in numerous studies where manipulating visual landmarks directly affects the firing of hippocampal spatial cells (O'Keefe & Conway, 1978; Jeffery & O'Keefe, 1999; Acharya et al., 2016). Indeed, place cells remap in darkness, and grid cells lose their characteristic hexagonal firing patterns without visual inputs (Chen et al., 2016; Pérez-Escobar et al., 2016; Waaga et al., 2022), highlighting vision's role in maintaining stable spatial representations while accommodating the integration of other sensory modalities.

Recent computational approaches have demonstrated considerable success in modelling this ego-to-allocentric transformation using recurrent neural networks (RNNs) trained on self-supervised predictive tasks. These models spontaneously develop spatially tuned units reminiscent of biological spatial cells. Various implementations have explored different aspects of spatial learning,

including reinforcement learning frameworks (Stachenfeld et al., 2017), navigation through abstract spaces (Whittington et al., 2020), prediction from visual information (Uria et al., 2022; Levenstein et al., 2024; Gornet & Thomson, 2024), integration of vestibular inputs (Cueva & Wei, 2018), or a sparse combination of multiple sensory modalities (Recanatesi et al., 2021). Moreover, Cueva & Wei (2018) examined the emergence order of spatial cells across training epochs, observing head direction cells first, then border cells, and finally grid cells – in rough accordance with experimental findings. These studies suggest that predictive learning – anticipating future sensory inputs based on current and past observations – may be a fundamental computational principle underlying the emergence of spatial representations in both artificial systems and biological brains.

While previous models have demonstrated that spatial representations can emerge from predictive learning, they have not addressed how developmental progression in sensorimotor experience influences spatial coding ontogeny. In this paper, we provide a computational framework for understanding how developmental changes in locomotion shape spatial representations in the hippocampus. We first classify locomotor development through computational analysis of experimental data, revealing distinct developmental stages. Using RNNs trained to predict visual inputs from concurrent visual and vestibular information, we demonstrate that developmental changes in movement patterns drive the emergence of spatially tuned units that recapitulate biological developmental timelines. Importantly, our model makes a novel prediction, namely the appearance of directional selectivity in place cells – a finding we validate through analysis of previously collected hippocampal recordings from developing rats.

## 2 COMPUTATIONAL ANALYSIS OF POSTNATAL LOCOMOTOR DEVELOPMENT

Altman & Sudarshan (1975) identified clear locomotion stages in rats through manual labelling of movement speed during open field foraging. We introduce a computational method to characterise developmental locomotor stages (briefly described hereafter, details in Appendix C). We aggregated data from multiple previous studies (Wills et al., 2010; Tan et al., 2015; Muessig et al., 2015; Bassett et al., 2018; Muessig et al., 2019) where rats explored familiar square open field environments.

For rats aged P11 to P25, we computed probability density functions of speed and rotational speed, along with transition probabilities between spatial bins. We quantified locomotor differences across ages using Jensen-Shannon distances between these distributions and transition matrices, then aggregated results into a correlation matrix (Figure 1a). Analysing individual rats rather than age-grouped cohorts captured individual variability in locomotor maturation – allowing rats of the same age to cluster differently based on their developmental stage. Two dimensional t-SNE visualisation (Van der Maaten & Hinton, 2008) of the correlation matrix revealed distinct clusters of movement statistics (Figure 1b). Gaussian mixture modelling identified three optimal clusters through minimising the Bayesian information criterion (Schwarz, 1978) (Figure 1c), corresponding to rats with median ages of P13.5, P16, and P20. We designate these developmental stages as crawl, walk, and run, respectively. An additional group included adult rats aged 3-6 months. The movement patterns characterised by each group provide the basis for generating synthetic trajectories used to train separate RNNs at each developmental stage.

### 2.1 SIMULATION OF NOVEL TRAJECTORIES CORRESPONDING TO DEVELOPMENTAL LOCOMOTION STAGES

We simulated novel rodent trajectories using the open-source toolbox RatInABox (George et al., 2024). The software implements an Ornstein-Uhlenbeck process – a continuous random walk with tendency to return to a central drift value. We tuned key parameters to match each of the three developmental locomotion clusters and the adult one observed experimentally: the mode of the Rayleigh distribution from which agent speed was sampled, the timescale over which speed decorrelated, the standard deviation of rotational velocity, and thigmotaxis strength – the tendency to stay near walls. Simulations generated agent position, head direction, speed, and rotational speed at 1 Hz temporal resolution (Figure 1e). We designed a landmark-rich virtual arena to replicate experimental conditions (Figure 1d). To emulate rat vision (Hughes, 1979), we captured low-resolution black and white frames (32 ×16 pixels) using a panoramic camera – with 240° horizontal and 120° vertical fields of view – positioned according to the agent's location and head direction (Figure 1f).

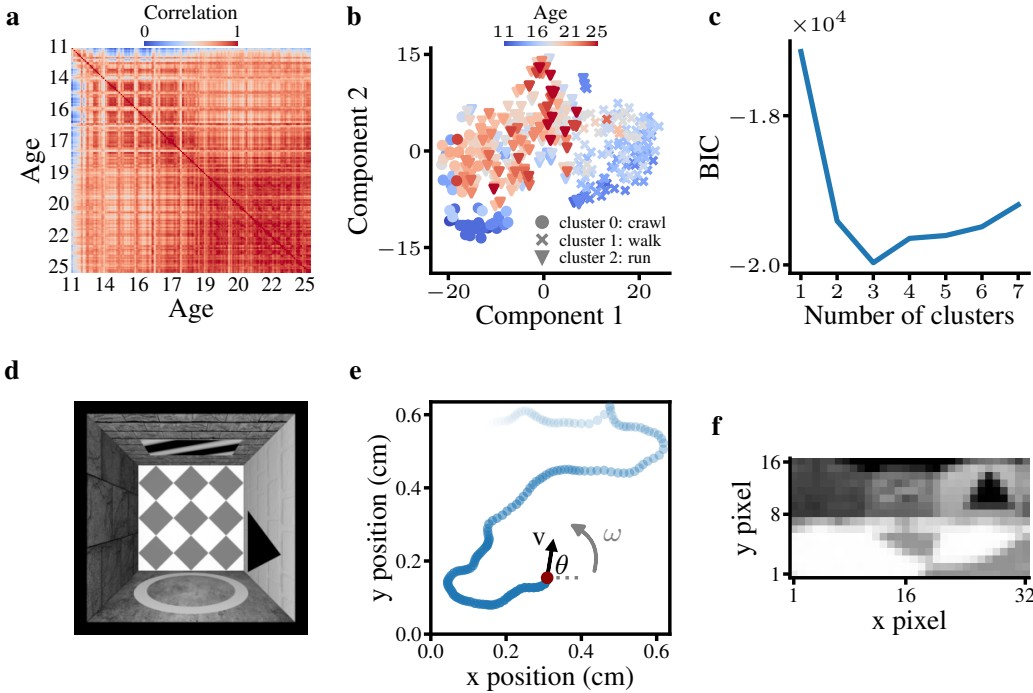

Figure 1: Clustering analysis of developmental locomotion stages and trajectory simulation. **a** Correlation matrix of locomotion metrics across rats aged P11-P25 (cold/warm colours indicate low/high correlations). **b** t-SNE visualisation of the correlation matrix. Marker shapes indicate locomotion clusters, colours show rat ages. **c** Bayesian information criterion (BIC) values for Gaussian mixture model with varying cluster numbers (lower values indicate better fit, minimum at 3 clusters). **d** Top view of the virtual arena with landmarks, used for simulations. **e** Example simulated trajectory showing agent's past positions (blue, fading with time), current position (red), head direction $\theta$, velocity $\mathbf{v}$, and rotational velocity $\omega$. **f** Visual frame showing the simulated rat's eye view (panoramic camera).

## 3 RESULTS

We used a single-layer recurrent neural network (RNN) as our artificial hippocampus model (Figure 2a). This network acts as a functional approximation of the hippocampal formation as a whole – it captures the fundamental computational principle of recurrent dynamics enabling the maintenance of internal states that predict future sensory input, and it is not intended as a literal circuit model of any specific subregion. The recurrence in our model abstracts over multiple sources of temporal integration in the biological system, including CA3 recurrent collaterals, the hippocampal-entorhinal loop, and broader cortico-hippocampal interactions. Moreover, our framework assumes inductive biases analogous to genetic programs through its architecture, input format, and learning objective. Indeed, hippocampal pyramidal neurons are generated between embryonic days (E) 16-21 (Bayer, 1980), establishing the architecture upon which experience operates.

The network was composed of $N_h = 500$ hidden units and it was trained to predict upcoming visual stimuli from concurrent visual and vestibular inputs. To emulate the information available to navigating rats, at each timestep $t$, the flattened frame $Y_t \in \mathbb{R}^{N_f}$ (where $N_f = 16 \cdot 32$ is the number of pixels in a frame), the velocity vector (x and y components) $\mathbf{v}_t$, and the rotational speed $\omega_t$ were concatenated to form the input $X_t = \text{concat}(Y_t, \mathbf{v}_t, \omega_t) \in \mathbb{R}^{(N_f+3)}$. The hidden state $H_{t+1} \in \mathbb{R}^{N_h}$ and the predicted frame $\tilde{Y}_{t+1}$ were computed as:

$$H_{t+1} = \sigma\left(X_t W_x^T + H_t W_h^T\right)$$
$$\tilde{Y}_{t+1} = H_{t+1} W_o^T \tag{1}$$

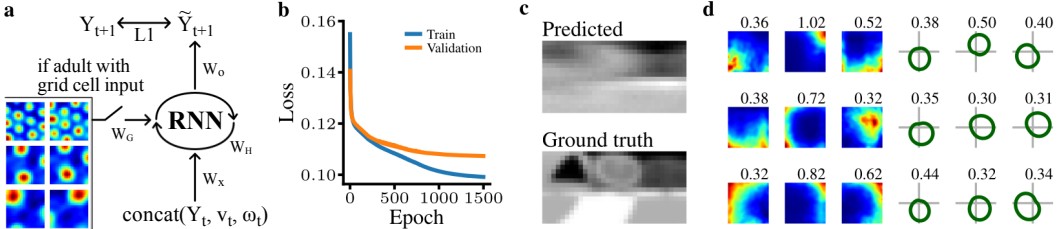

Figure 2: Recurrent neural network (RNN) model of hippocampal function. **a** RNN schematic. At each timestep $t$, the network received a concatenation of the agent's visual input $Y_t$, velocity vector $\mathbf{v}_t$, and rotational velocity $\omega_t$. For the adult locomotion stage, an additional variant was trained – where the hidden layer was initialised to a linear projection $W_g$ of a population of grid cells at trajectory onset and every 90 timesteps. The predicted frame $\tilde{Y}_{t+1}$ is decoded from the hidden state with a linear projection $W_o$ and compared to the observed frame $Y_{t+1}$ using the L1 distance. **b,c,d** RNN trained on simulated trajectories belonging to the adult locomotion stage. **b** Train (blue) and validation (orange) loss trends throughout training. **c** Example predicted frame (top) and its ground truth comparison (bottom), sampled from the validation set. **d** Subset of rate maps in (x,y) space (warmer colour indicate higher activity) of hidden units classified as place cells and corresponding $\mathrm{SI}_r$ values (left), and subset of polar maps in polar space of hidden units classified as head direction cells and corresponding $\mathrm{SI}_d$ values (right).

Where $\sigma$ is the sigmoid function, and $W_x \in \mathbb{R}^{(N_f+3) \times N_h}$, $W_h \in \mathbb{R}^{N_h \times N_h}$, and $W_o \in \mathbb{R}^{N_h \times (N_f+3)}$ are the learnable weight matrices applied to the input, the previous hidden state, and the current hidden state, respectively. The loss was calculated as the average absolute distance between the predicted frame $\tilde{Y}_{t+1}$ and the ground truth $Y_{t+1}$.

We trained the network on movement data corresponding to each developmental stage (Section 2), beginning with the youngest rats and continuing through maturation. To emulate continuous learning during development, each network inherited weights from the previous stage. Namely, we trained networks sequentially: starting from scratch on crawl (youngest), then fine-tuning on walk, run, and finally on adult. For each locomotion stage, we trained models on 17 sequences of 720 frames (12,240 samples total, equivalent to 3.5 hours of simulated experience) of simulated trajectories (Section 2.1). We employed back-propagation through time with teacher forcing on 9-timestep windows, and trained for 1,500 epochs – which was enough to reach convergence (Figure 2b). Validation of each model used 4 simulated sequences (2,880 frames, 1 hour of experience).

When training on simulations of developing rats, the hidden layer was simply initialised to zero at the start of each trajectory. For the adult locomotion stage specifically – to investigate the functional role of grid cells – we also trained a variant of our RNN which included grid cell input. This addition is justified by previous studies showing grid cells mature only in adulthood (Wills et al., 2010; Bjerknes et al., 2014). In this variant, we trained the network from scratch and initialised the hidden layer at trajectory onset and every 90 timesteps (1.5 minutes) to a linear learnable projection of 25 grid cells $G_t$ with varying grid scales (specifying firing field spacing and size):

$$H_t = \begin{cases} G_t W_g^T & \text{when } t\%90 = 0 \\ \text{Eq. 1} & \text{otherwise} \end{cases} \tag{2}$$

We analysed hidden unit activity with respect to agent position and head direction using 9 test sequences (3,600 frames, 2 hours of experience). To quantify spatial selectivity, we computed rate maps – two-dimensional representations of neural firing rates across spatial locations. The use of rate maps is standard practice in neuroscience for analysing place cell recordings (Muller & Kubie, 1989). We divided the arena into a 25×25 grid and calculated rate map $R$ of unit $k$ by averaging its activity within each spatial bin:

$$R_{i,j}^{(k)} = \frac{\sum_t A_t^{(k)} \cdot \mathbb{1}_{\mathbf{P}_t \in \mathrm{Bin}_{(i,j)}}}{\sum_t \mathbb{1}_{\mathbf{P}_t \in \mathrm{Bin}_{(i,j)}}} \quad i,j \in \{1,\ldots,B_p\} \times \{1,\ldots,B_p\} \tag{3}$$

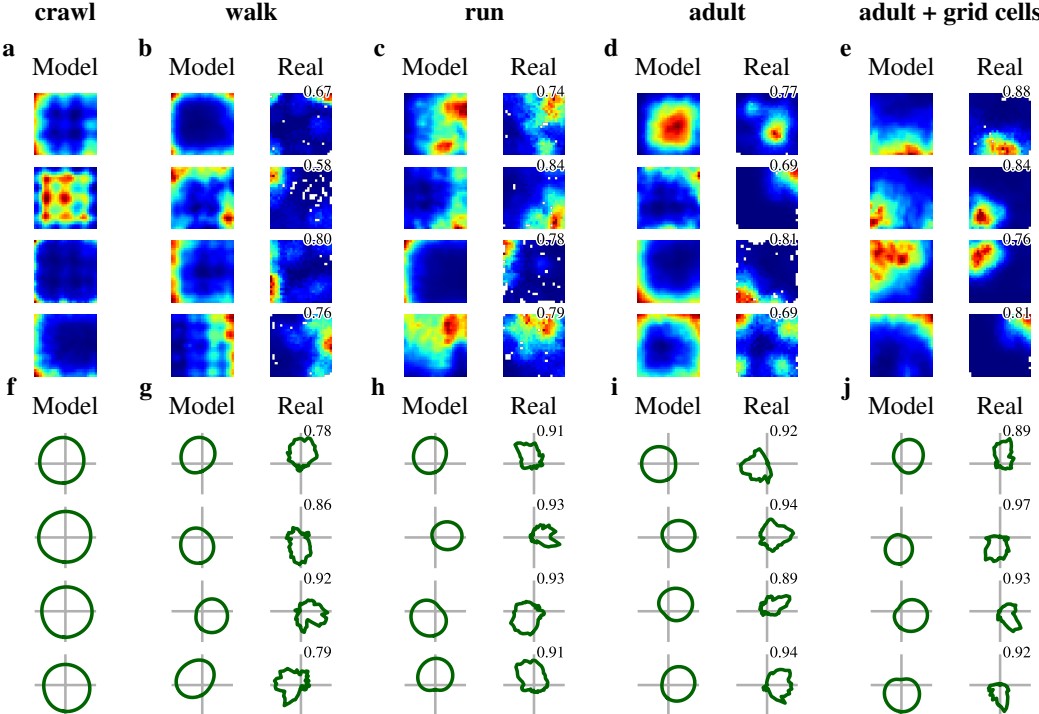

Figure 3: Randomly selected rate maps (**a-e**) and polar maps (**f-j**) of units classified as place cells or head direction cells from RNNs trained at the different locomotion stages (Model), and corresponding neurons with the highest correlation (shown on the top right of each matched cell) from hippocampal CA1 recordings (Real). Rate maps from **a** crawl (no comparison with real data), **b** walk, **c** run, **d** adult, and **e** adult with grid cell input. Polar maps from **f** crawl (no comparison with real data), **g** walk, **h** run, **i** adult, and **j** adult with grid cell input.

where $A^{(k)}$ is the activity of hidden unit $k$, $\mathbf{p}$ is the agent's position vector (x and y components), and $B_p$ is the number of bins along each direction. Both numerator and denominator were smoothed using a Gaussian filter ($\sigma = 0.75$ bins). We quantified spatial selectivity using spatial information ($\text{SI}_r$, definition in Appendix E), computed on min-max normalised rate maps – following the standard approach of subtracting the minimum and dividing by the range (difference between maximum and minimum values).

For directional selectivity, we calculated rate maps in polar space (polar maps) by binning head direction into 60 angular bins. We measured directional tuning using the resultant vector length – which quantifies the concentration of the directional preference – and spatial information of a polar map ($\text{SI}_d$), both defined in Appendix E. Following standard neuroscience classification criteria (Muller & Kubie, 1989; Taube et al., 1990), units with $\text{SI}_r > 0.3$ were classified as place cells, and units with resultant vector length $> 0.3$ or $\text{SI}_d > 0.2$ as head direction cells.

## 3.1 MATURATION OF LOCOMOTION EXPLAINS HIPPOCAMPAL SPATIAL CELL DEVELOPMENT

When training completed, the RNN successfully learned the visual prediction task, as evidenced by decreasing validation loss and accurate frame reconstructions (Figure 2b,c). When trained with adult locomotion, hidden units developed clear spatial and directional tuning (Figure 2d). We next examined how spatial representations emerge through locomotor development, extracting latent representations from models trained on each stage. We compared our networks' hidden state to hippocampal CA1 recordings (Wills et al., 2010; Muessig et al., 2015; 2019) from developing rats (P14-P25) and adults during open-field foraging – Figure 3 shows a qualitative comparison of a subset of neurons. These experimental trajectories were part of the data used in our clustering analysis (Section 2). No rat with hippocampal recording was part of the crawl stage, a consequence of the fact that younger

rats do not have sufficient position coverage to asses two dimensional firing correlates. Our comparisons focused on CA1 because this is the region for which developmental data was available and – as the anatomical output of the hippocampal formation – CA1 responses are indicative of the system's final processed output.

We quantified spatial selectivity by computing spatial information from rate maps ($\text{SI}_r$) of units in the network's latent space and of hippocampal neurons (Figure 4b). For directional selectivity, we calculated spatial information ($\text{SI}_d$) and resultant vector length from polar maps of artificial and biological neurons (Figure 4d,e). Experimental data showed significantly increasing trends across all spatial metrics throughout development, confirmed by Jonckheere-Terpstra (JT) tests for ordered differences ($p < 0.001$ for all metrics). Similarly, the model showed – from walk to adult – significantly increasing trends in spatial tuning (JT test: $\text{SI}_r$ $p < 0.01$, directional metrics $p < 0.001$). We validated model-experiment correspondence by comparing Jensen-Shannon distances between model and experimental distributions against null distributions generated by shuffling network units across stages 10,000 times (Figure 4f). Using established neuroscience classification criteria (Muller & Kubie, 1989; Taube et al., 1990), we classified network units and hippocampal neurons as place cells (Figure 4a) or head direction cells (Figure 4c). Experimental data showed significant increases in both cell types across locomotion stages (JT test: place cells $p < 0.001$, head direction cells $p < 0.05$). Networks trained sequentially across locomotion clusters recapitulated the progressive emergence of spatial tuning observed in hippocampal development (Figure 4), with one important exception: purely spatial place cells maturation required grid cell input in the adult model to reach experimentally observed levels. This highlights that mature positional tuning, and consequently the full increase in place cell numbers, may depend on grid cell input in adult rats.

These results prove that sensory experiences shaped by movement play a key role in spatial representation emergence and maturation. Additional supporting evidence includes decreased decoding errors for position and head direction from hidden-layer activity (Figure 5a), indicating formation of an allocentric map of the environment. To quantify the functional importance of spatial cells (Luo et al., 2024), we performed lesion studies by silencing classified place or head direction cells and measuring validation loss. Across 100 repetitions per stage, spatial cells became increasingly critical for task performance (Figure 5b), confirming their role in spatial knowledge acquisition. Finally, we assessed the robustness of spatial representation emergence by calculating inter-trial correlations, and training RNNs with different parameters. Population-level Pearson correlation analyses between rate maps (and polar rate maps) computed on the first and second halves of the held-out validation trajectories consistently remained above 0.8 across development (Figure A8). Moreover, reducing or increasing by 25% the number of hidden units, training for up to 2,500 epochs at each stage, and halving or doubling the input dimensions did not alter spatial tuning maturation (Figures A3 and A4). These additional results demonstrate the stability of our findings.

## 3.2 THE MODEL SUGGESTS A DEVELOPMENTAL TREND IN THE DIRECTIONAL SELECTIVITY OF PLACE CELLS VALIDATED BY REAL DATA

The observed increase in directional tuning and head direction (HD) cell numbers in our model (Figure 4c-e) occurs primarily in neurons that also encode position (Figure 4g, solid line), while pure directional selectivity emerges at the "walk" stage – corresponding to approximately P16, with instances at younger ages – and remains stable thereafter (Figure A9). This pattern of pure directional tuning aligns with experimental findings reporting adult-like HD cells by P15 in dorsal presubiculum and medial entorhinal cortex (Wills et al., 2010; Langston et al., 2010), with immature HD signals observable as early as P12 (Tan et al., 2015; Bjerknes et al., 2015). However, the emergence of mixed selectivity in spatial neurons, and specifically the development of directional selectivity in CA1 neurons, has not been previously studied.

To investigate whether the emergence of conjunctive place-HD tuning suggested by our model is present in experimental hippocampal data, we applied corrected polar maps (Burgess et al., 2005) following the distributive hypothesis (Muller et al., 1994) to control for potential confounds such as inhomogeneous orientation sampling. We classified cells as HD cells only when both standard criteria ($\text{SI}_d$ and resultant vector length) and corrected polar map criteria were met, with Pearson correlation between original and corrected maps exceeding 0.5 – identifying correlations below this threshold as sampling artefacts. This analysis revealed a significant, previously unreported developmental increase in conjunctive place-direction cells (Figure 4g, dashed line) in hippocampal

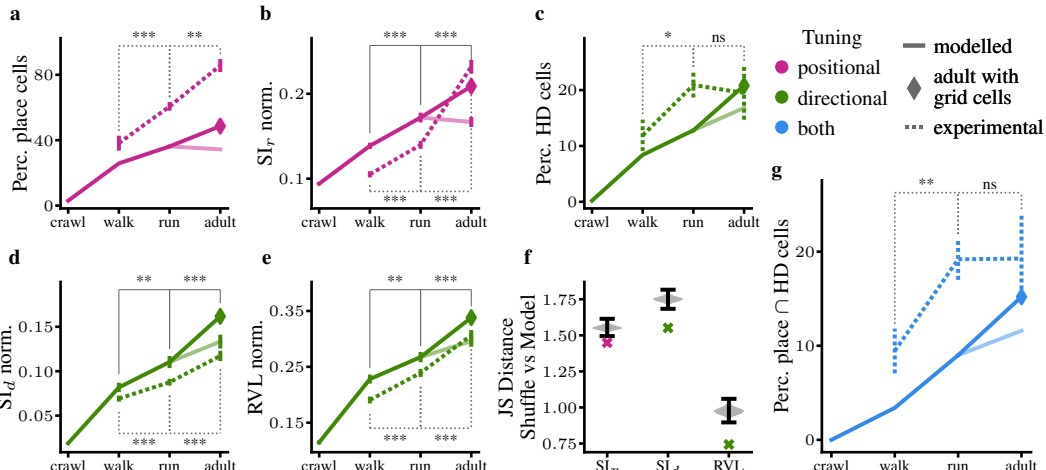

Figure 4: Maturation of locomotion explains the development of hippocampal spatial cells. We refer to RNNs with the locomotion cluster their training data was simulated from. Experimental data (dotted lines) shows simultaneously recorded ensembles of cells, grouped by the rat's locomotion stage at the age of recording. Diamonds indicate adult model trained with grid cell input. Asterisks denote significance levels from one-sided pairwise Wilcoxon rank-sum tests (Wilcoxon, 1992) with Benjamini-Hochberg correction (Hollander et al., 2013) testing for increasing differences (exact p-values in Table A4). **a** Percentage of RNN units (solid line) and percentage (mean±SEM) of hippocampal neurons (dotted line) classified as place cells. **b** Min-max normalised spatial information $SI_r$ (mean±SEM) for rate maps of RNN units (solid) and hippocampal neurons (dotted). **c** Percentage of RNN units (solid) and percentage (mean±SEM) of hippocampal neurons (dotted) classified as head direction (HD) cells. **d** Min-max normalised spatial information $SI_d$ (mean±SEM) for polar maps of RNN units (solid) and hippocampal neurons (dotted). **e** Min-max normalised resultant vector length (RVL, mean±SEM) for polar maps of RNN units (solid) and hippocampal neurons (dotted). **f** Comparison of the overall Jensen-Shannon (JS) distance (crosses) – calculated between model (no grid cell input) and experimental distributions for metrics shown in **b**, **d**, **e** – with null distributions generated by shuffling network units across clusters 10,000 times (violin plots). **g** Percentage of RNN units (solid) and percentage (mean±SEM) of hippocampal neurons classified as both place and HD cells (dotted).

recordings (JT test, p < 0.01). Thus, our model prediction led to a new experimental finding: gradual increases in directionality within hippocampus occur primarily through conjunctive place-HD cells rather than pure HD cells. Furthermore, in adult rats, our results confirmed Acharya et al. (2016)'s finding that approximately 25% of CA1 pyramidal neurons exhibit significant directional selectivity – in both real-world and virtual environments. Similar levels of directional tuning have been reported in bats (Rubin et al., 2014) and mice (Jercog et al., 2019), suggesting this is a conserved feature across mammalian species.

## 3.3 SPATIAL CELL DEVELOPMENT IS NOT EXPLAINED BY RATE OF CHANGE IN SENSORY STIMULI, BEHAVIOURAL REPERTOIRE, OR AMOUNT OF TRAINING DATA

Levenstein et al. (2024) demonstrated that RNNs trained to predict visual inputs over extended spatio-temporal sequences develop neural manifolds reflecting environmental structure. They visualised these manifolds using Isomap projection and assessed them with spatial Representation Similarity Analysis (sRSA) (Levenstein et al., 2024), which quantifies correspondence between neural and spatial distances. This suggests a transition from egocentric (reconstructive) to allocentric (predictive) spatial representations. Given that locomotor development involves increasing speed, our task naturally becomes more predictive as inter-frame distances increase (Figure 5d, black line). This transition is reflected in our network's sRSA trend (Figure 5c) and Isomap plots (Figure 5e).

To test whether increased predictive demands are sufficient to drive allocentric representations, we trained networks on crawl trajectories with extended inter-frame intervals (1-3 seconds), achieving

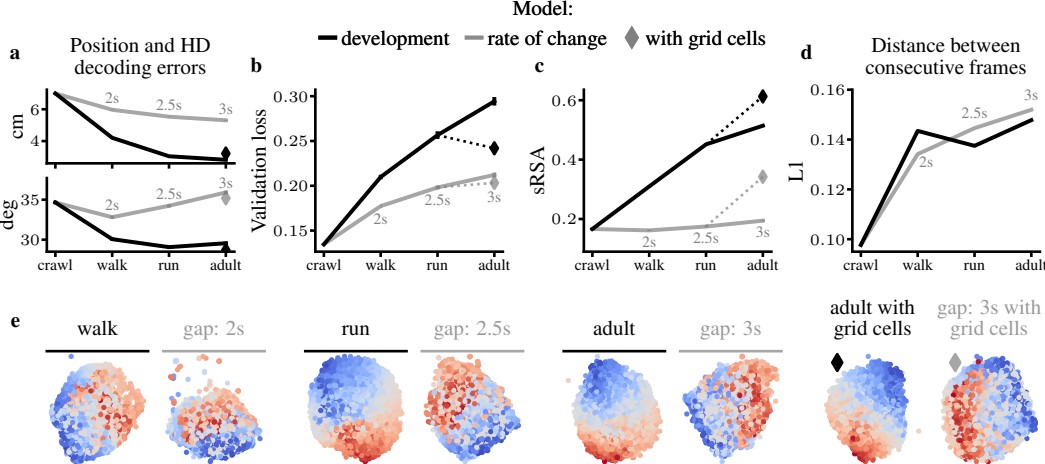

Figure 5: Behaviour-driven sensory experience – not rate of change alone – drives the emergence of allocentric spatial representations. Black lines show RNNs trained on trajectories modelled from developmental stages, grey lines show RNNs trained on crawl trajectories with extended sampling gaps (labelled in each plot), diamonds indicate models trained with grid cell input. **a** Position (top) and head direction (bottom) decoding errors (mean±SEM) using linear regression on hidden units (50-50 train-test split on validation data). **b** Validation loss (mean±SEM) when silencing equal numbers of classified place or HD cells (100 repetitions). **c** Spatial representational similarity analysis (mean±SEM) quantifying correlation between agent position and hidden unit activity. **d** L1 distance between consecutive frames (mean±SEM) in validation set. **e** Isomap visualisation of hidden unit activity on random trajectory excerpts, coloured by agent position (x + y coordinates).

distances comparable to adult stage (Figure 5d). However, this manipulation failed to reproduce the developmental phenomena observed in experimental data (Figure A5). In fact, the original models were significantly more likely to have generated the experimental data, as shown by comparing the Bayesian information criterion ($\Delta$BIC > 10; likelihood calculations are described in Appendix F). Crucially, networks trained with extended intervals had less informative spatial responses compared to those trained on naturalistic developmental trajectories, as evidenced by higher decoding errors (Figure 5a), reduced importance in lesion studies (Figure 5b), weaker sRSA (Figure 5c), and less structured Isomap projections (Figure 5e). Even with grid cell inputs, these models were less likely to have generated the experimental data ($\Delta$BIC > 10) and achieved lower spatial encoding than adult networks (Figure 5), confirming that behavioural changes beyond temporal sampling drive spatial representation development.

Finally, we compared our model's results to two control experiments that isolate different factors in spatial representation development. (i) We trained RNNs on reversed locomotion development (from adult to crawl), which immediately developed robust spatial representations (Figure A5), accurate decoding and strong sRSA (Figure A6). Subsequent training on simpler movement patterns produced fewer changes in spatial tuning than our developmental model. (ii) We trained networks exclusively on crawl movement, matching the total training exposure of the original model, which failed to produce spatial representations, with minimal changes in spatial tuning (Figure A5), decoding accuracy, and sRSA (Figure A6). Together, these experiments demonstrate that the developmental progression is not merely about behavioural repertoire or accumulated training; rather, it is the specific trajectory of increasing movement complexity that drives spatial representation emergence.

## 4 DISCUSSION

Our work establishes a mechanistic link between locomotor development and the emergence of spatial representations in the hippocampus. Through a computational classification of movement trajectories from experimental data (Wills et al., 2010; Tan et al., 2015; Muessig et al., 2015; Bassett et al., 2018; Muessig et al., 2019), we identify three distinct stages of locomotor development: crawl

(median age P13.5), walk (P16), and run (P20), plus adult locomotion. We introduce a shallow RNN model of the hippocampus that – when sequentially trained on movement patterns from these stages – recapitulates the developmental timeline of spatial cells observed in recordings of hippocampal cells of maturing rats (Wills et al., 2010; Muessig et al., 2015; 2019). This demonstrates that sensory experiences shaped by locomotion are fundamental to the formation of a cognitive map of space. More broadly, our results underscore how movement kinematics critically influence the emergence of spatial representations in self-supervised neural networks trained on egocentric sensory inputs.

A particularly novel prediction arising from our model is that directional selectivity in the hippocampus emerges primarily through conjunctive place-direction coding rather than in pure head direction cells. We investigated this prediction in CA1 recordings and confirmed a significant developmental increase in cells classified as both place and head direction cells – a pattern not previously highlighted in developmental studies. This observation aligns with previous work on adult animals across species, where approximately a quarter of place cells exhibit significant directional modulation (Rubin et al., 2014; Acharya et al., 2016; Jercog et al., 2019). The developmental trajectory of this conjunctive coding may reflect the increasing computational demands of navigation as locomotion becomes more complex.

Our controlled experiments dissociating rate of change in visual input from actual locomotion improvements demonstrate that spatial representation emergence cannot be explained by increasing temporal gaps between sensory experiences alone. Even when RNNs trained on early-stage locomotion patterns with extended sampling intervals (creating a prediction task comparable to the adult one), they fail to develop the same level of spatial representations observed in models trained on sequences of trajectories simulated from rat development. This result suggests the specific statistics of movement are critical for the emergence of an allocentric view of space.

Our results demonstrate that within a fixed network architecture – assuming inductive biases analogous to genetic programs – the specific temporal structure of locomotor experience accounts for key phenomena of spatial cell maturation. This complements recent studies showing that birthdate-dependent preconfiguration interacts with experience-dependent refinement (Cavalieri et al., 2021; Huszár et al., 2022) and that rearing environments shape developing spatial representations (Ulsaker-Janke et al., 2023; Farooq & Dragoi, 2024). Together, our findings establish locomotor statistics as a critical and previously unexamined developmental factor and inform the design of biologically-inspired navigation models.

However, several limitations warrant discussion. Our simplified visual and vestibular inputs do not fully replicate the rich multisensory environment of developing rats – future work incorporating olfactory, somatosensory, and auditory modalities could provide more comprehensive insights (Wills et al., 2014; Guthrie & Gall, 2003; Grant et al., 2012). Additionally, while we quantitatively compare our model to experimental recordings of CA1, spatial development involves complex interactions across brain regions. Finally, we simplified vestibular and grid cell systems as velocity inputs and initialisation signals, respectively. Achieving adult-like positional tuning required the RNN to be trained from scratch with grid cell injections. This highlights incomplete understanding of how different brain regions coordinate during spatial system maturation. The rapid timescale of grid cell maturation (Wills et al., 2012) hints that grid cells may rely more heavily on intrinsic circuit mechanisms – possibly including attractor-like dynamics within medial entorhinal cortex (Burak & Fiete, 2009; Couey et al., 2013), rather than purely experience-dependent learning. While this distinct developmental profile supports our treatment of grid cell input as an external signal, the precise mechanisms and potential interactions during this transition period could be addressed in further research.

Future work could explore several other promising directions. Examining how locomotor development interacts with environmental or sensory manipulations would extend our framework to test computational predictions against recent empirical works documenting the effect of rearing environment on spatial representation emergence in the developing hippocampus (Ulsaker-Janke et al., 2023; Farooq & Dragoi, 2024). Furthermore, investigating the developmental trajectory of border cells across locomotion stages represents another promising path for extending this work and possibly corroborating previous experimental findings of border cell emergence in development (Bjerknes et al., 2014). To conclude, we suggest future experimental manipulations of locomotor development could directly test our key prediction about conjunctive place-direction cell emergence, potentially through targeted interventions during critical developmental periods.

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

APPENDIX

## A LLM USAGE STATEMENT

Large language models (LLMs) were employed exclusively to improve the quality of written English and for stylistic refinement, with attentive supervision by the authors. No scientific content, experimental design, data analysis, or any intellectual contribution were generated through LLMs, nor did such assistance influence any reported results.

## B EXPERIMENTAL DATASETS

In our work, we aggregated experimental data from multiple previous studies (Wills et al., 2010; Tan et al., 2015; Muessig et al., 2015; Bassett et al., 2018; Muessig et al., 2019), where rats explored a familiar square open field environment. All data was included in the locomotor development clustering analysis of Section 2, while a subset – with recordings from hippocampal area CA1 (Wills et al., 2010; Muessig et al., 2015; 2019) – was used for comparison with results from modelling of Section 3.1. Experiments were performed on P11 to P25 (and adults) Lister Hooded rats (Figure A1a) in a familiar square-walled (62.5 cm sides, 50 cm high) light-gray wooden box, placed on a black, square platform – while rats searched for drops of soya-based infant formula milk randomly scattered on the floor of the environment. Behavioural tests began at different ages for different rats, hence experience of the arena was dissociated from age. For specific information about experimental settings, we redirect the reader to the relevant publications.

## C LOCOMOTOR DEVELOPMENT CLUSTERING ANALYSIS

### C.1 METRICS EXTRACTION

In the clustering analysis, we included only experimental trials lasting 300 seconds or more. Trials exceeding 900 seconds were truncated to this duration. For rats ages P11 to P25, a maximum of 60 trials were selected at random. We estimated metrics for each rat at each age. Probability density functions of speed were computed with histograms of 50 bins in the range 0 to 0.15 m/s. Similarly, probability density functions of rotational speed were generated with 50 bins spanning -360 to 360 °/s. Transition probabilities were calculated by constructing a transition matrix $TM$ that encodes the likelihood of moving between spatial bins within the environment. The arena was divided into a 10x10 grid ($B_T = 100$ bins). $TM \in \mathbb{R}^{B_T \times B_T}$ was initialised to a matrix of zeros and – for each position the rat visited $\mathbf{p}_t = (x_t, y_t)$, belonging to bin $b_t$ – it was updated as:

$$TM_{(b_t, b_{t-k})} = TM_{(b_t, b_{t-k})} + (0.5)^{\frac{k-1}{2}}, \quad k = 1 \rightarrow K \tag{4}$$

where $K$ was set to 7 seconds and indicates the threshold over which accumulating the (discounted) transitions, and $t \geq K$. The value in $TM$ corresponding to the last bin visited ($k = 1$) before reaching $b_t$ was incremented by 1, while the entry of $TM$ corresponding to the bin before that ($k = 2$) was incremented by around 0.71, and so on. At the end of a trial, $TM$ was normalized by the occupancy matrix – which counted the number of times each bin was visited – and smoothed using a Gaussian filter ($\sigma$=1 bin).

### C.2 CLUSTERING OF POSTNATAL LOCOMOTOR DEVELOPMENT

Clustering was used to group developmental stages with similar locomotor patterns based on the metrics introduced in Section C.1 – specifically, probability density functions of speed and rotational speed, as well as transition matrices. We quantified locomotor differences using Jensen-Shannon (JS) distances between the distributions and between the transition matrices. The JS distance between two vectors $\mathbf{p}$ and $\mathbf{q}$ was defined as:

$$\text{JS}(\mathbf{p}, \mathbf{q}) = \sqrt{\frac{\text{KL}(\mathbf{p}||\mathbf{m}) + \text{KL}(\mathbf{q}||\mathbf{m})}{2}} \tag{5}$$

where $\mathbf{m}$ is the pointwise mean of $\mathbf{p}$ and $\mathbf{q}$, and KL is the Kullback-Leibler divergence. When comparing transition matrices, JS distances were calculated over the first dimension and averaged over the second.

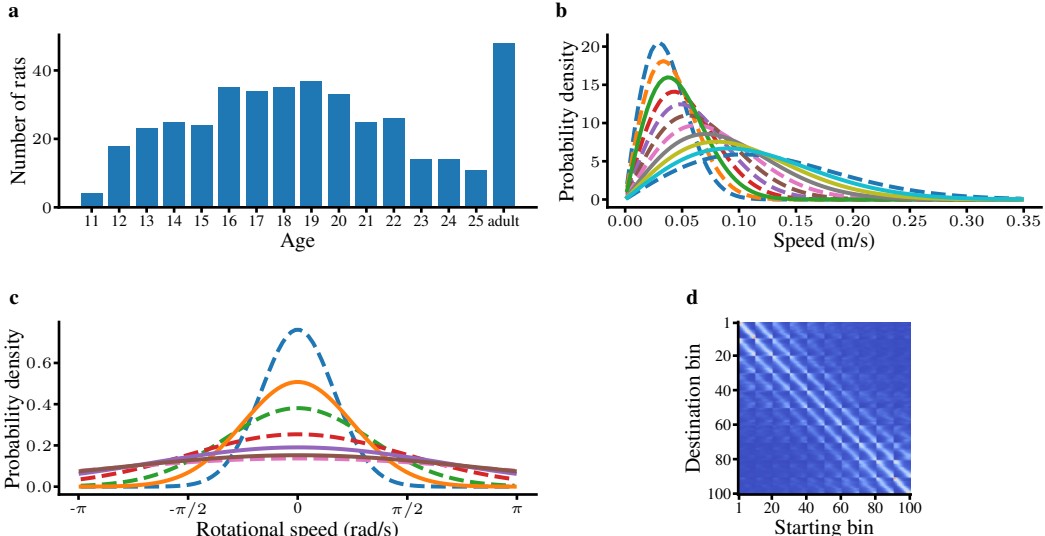

Figure A1: Simulation of new trajectories emulating experimental data. **a** Histogram of ages of rats used in the clustering analysis. Adult rats are aged 3-6 months. **b** Speed probability density functions of trajectories simulated for the grid search. Solid lines indicate curves from selected parameters. **c** Rotational speed probability density functions of trajectories simulated for the grid search. Solid lines indicate curves from selected parameters. **d** Transition matrix estimated from adult trajectories. Warmer colours indicate higher probability to move from the starting bin to the destination bin.

More formally, given a sample $r \in R$ where $R$ is the set of rats at all ages, a probability density function of speed $P_S^{(r)}$, of rotational speed $P_\omega^{(r)}$, and a transition matrix $TM^{(r)}$ were calculated. Then, these metrics were compared to all other rats $r' \in R \quad \{r \neq r'\}$ using the JS distance in order to construct a distance matrix

$$
\begin{aligned}
D_{(r,r')}^{(P_S)} &= \text{JS}\left(P_S^{(r)}, P_S^{(r')}\right) \\
D_{(r,r')}^{(P_\omega)} &= \text{JS}\left(P_\omega^{(r)}, P_\omega^{(r')}\right) \\
D_{(r,r')}^{(TM)} &= \frac{1}{B_T} \sum_{b \in B_T} \text{JS}\left(TM_{(:,b)}^{(r)}, TM_{(:,b)}^{(r')}\right)
\end{aligned}
\tag{6}
$$

Distances were normalised and aggregated into a single distance matrix $D \in \mathbb{R}^{|R| \times |R|}$ as

$$
D = \sum_{X \in \{P_S, P_\omega, TM\}} \frac{D^{(X)}}{\max D^{(X)}}
\tag{7}
$$

A radial basis function kernel matrix $K$ was derived from $D$ as

$$
K_{(r,r')} = \exp\left(-D_{(r,r')}^2\right)
\tag{8}
$$

where $\{r, r'\} \in R$. Kernel principal component analysis (Schölkopf et al., 1997) was applied to $K$ and the first $c = 15$ principal components were extracted. We estimated clusters from principal components by fitting a Gaussian mixture (GM) model. To determine the optimal number of clusters, the Bayesian information criterion (BIC) was minimized while varying the number of components $n$ (from 1 to 7) used for the GM model as

$$
\min_{n \in [1,7]} \text{BIC} = \min_{n \in [1,7]} \left(-2\,\overline{L}_n\,|R| + d(n)\,\ln(|R|)\right)
$$

$$
d(n) = \frac{n\,(c+1)\,c}{2}
\tag{9}
$$

Table A1: RatInABox agent parameters grid search

| Parameter | Description | Range |
|---|---|---|
| $\sigma_{\mathrm{v}}$ [m/s] | Scale parameter and mode of speed Rayleigh distribution. | `logspace(-5.1, -3.3, 11, base=2)` |
| $\tau_{\mathrm{v}}$ [s] | Timescale over which speed decorrelates. | `linspace(0.7, 2, 3)` |
| $\sigma_{\omega}$ [rad/s] | Std of rotational velocity Normal distribution. | `linspace(π/6, 6.5π/7, 7)` |
| thigmo | Bias towards following walls. | `[0.4, 0.5, 0.6]` |

Table A2: RatInABox selected parameters

| Locomotion stage | $\sigma_{\mathrm{v}}$ [m/s] | $\tau_{\mathrm{v}}$ [s] | $\sigma_{\omega}$ [rad/s] | Thigmo |
|---|---|---|---|---|
| crawl | 0.03794 | 2 | $\pi/4$ | 0.6 |
| walk | 0.09087 | 1.35 | $2\pi/3$ | 0.6 |
| run | 0.07081 | 1.35 | $2\pi/3$ | 0.6 |
| adult | 0.08021 | 1.35 | $2.5\pi/3$ | 0.4 |

where $\overline{L}_n$ is the per-sample average log-likelihood when setting $n$ components, and $d$ is the number of free parameters. Three locomotion clusters were found to minimize the BIC, which we designate crawl, walk, and run. For this analysis, the Python software libraries Scikit-learn, Scipy, and Pandas were used.

## C.3    RAT TRAJECTORIES SIMULATION

Rat trajectories were simulated using RatInABox (George et al., 2024), an open-source library which provides position, head direction, velocity, and rotational speed information (among other data) at a temporal resolution set by the user. The software implements two independent Ornstein-Uhlenbeck processes for speed and rotational speed. The rat moved in a squared arena with sides of length 62.5 cm. Data was collected at 50 Hz for the clustering analysis of Section 2 (matching experimental setup), and at 1 Hz for modelling results of Section 3.

To simulate locomotor development, a grid search over key RatInABox parameters was performed (Table A1) and the combinations optimally matching locomotion stages were selected. An additional parameter set was picked to match adult locomotion of rats aged 3-6 months. In more detail, for each set of parameters, we estimated probability density functions of speed, rotational speed, and a transition matrix (Figure A1b-d). For each locomotion stage, the set of parameters yielding simulated trajectories closest (in terms of JS distance) to experimental data was selected (Table A2).

## C.4    VIRTUAL REALITY

To recreate the visual environment, Blender was used to design a virtual arena that mimicked the experimental conditions. The virtual box featured brick-like textures on the walls, a rhomboid-patterned floor, and distinct visual landmarks attached to three of the walls. A camera was positioned 3.5 cm above the ground to collect visual frames according to the agent's location and head direction. The panoramic camera was configured to emulate rat's vision with a horizontal field of view of $240°$, a vertical field of view of $120°$, and a resolution of 32 by 16 pixels, capturing black-and-white images. Visual frames were rendered using the Blender Python API.

Table A3: RNNs training parameters

| Parameter | crawl | walk | run | adult | adult with grid cells |
|---|---|---|---|---|---|
| Hidden units $N_h$ | 500 | 500 | 500 | 500 | 500 |
| Initialisation | random | crawl | walk | run | random |
| Optimizer | RMSprop | RMSprop | RMSprop | RMSprop | RMSprop |
| Epochs | 1500 | 1500 | 1500 | 1500 | 1500 |
| Learning rate | 5e-5 | 5e-5 | 5e-5 | 5e-5 | 1e-4 |
| BPTT steps | 9 | 9 | 9 | 9 | 9 |
| Number of grid cells | - | - | - | - | 25 |
| Grid cells scale | - | - | - | - | 0.2, 0.4, 0.6 |
| Grid cells orientation | - | - | - | - | 0.1 |

Figure A2: Training and validation loss trends for models: **a** crawl, **b** walk, **c** adult, and **d** adult with grid cell input.

## D   RECURRENT NEURAL NETWORK

A single-layer recurrent neural network (RNN) was trained to predict upcoming visual stimuli from concurrent visual and vestibular inputs. Modelling was performed using Python and PyTorch, on a machine with a single CPU and single GPU (Nvidia RTX 4000 Ada Gen with 20 GB).

**Training**   Table A3 presents the training parameters used for each network. Biologically-plausible parameters that allow the network to reach convergence (validation loss plateau, Figure A2) were chosen without any parameter optimization. For the "rate of change" models introduced in Section 3.3, parameters were the same as for our original models, except for the inter-frame gap which was increased in a range from 1 to 3 seconds.

**RNN with grid cell input**   The hidden layer of the RNN with grid cell input trained on the adult trajectories was initialised to a linear projection of a population of 25 grid cells $G_t \in \mathbb{R}^{25}$ at the start of a trajectory and consequently every 90 timesteps. Given the wave orientation $\theta = 0.1$, a randomly sampled grid-scale $\lambda_i \in \{0.2, 0.4, 0.6\}$ and phase offset $\phi_i \sim \mathcal{U}(0, 2\pi)$, the firing rate of cell $i$ at time $t$ was defined as

$$G_t^{(i)} = \frac{1}{3} \max \left( 0, \sum_{a \in \{0, \pi/3, 2\pi/3\}} \cos \left( 2\pi \frac{\mathbf{p}_t \, \mathbf{e}_{\theta+a}}{\lambda_i} + \phi_i \right) \right) \tag{10}$$

where $\mathbf{p}_t = (x_t, y_t)$ is the position of the agent at time $t$ and $\mathbf{e}_\theta$ is the unit vector pointing toward direction $\theta$.

**Position and head direction decoding**   Position and head direction of the agent were decoded from the activity of hidden units. An ordinary least squared linear regressor was trained on half of the validation data and decoding errors were reported for the other half of the validation data. Given

predicted position $\tilde{\mathbf{p}}_t$ and head direction $\tilde{\theta}_t$, decoding errors were calculated as

$$
\begin{aligned}
\Delta \mathbf{p}_t &= |\mathbf{p}_t - \tilde{\mathbf{p}}_t|_2 \\
\Delta \theta_t &= |\theta_t - \tilde{\theta}_t|
\end{aligned}
\tag{11}
$$

**Lesion of hidden units**  Lesions were performed to investigate the functionality of hidden units with high spatial tuning with respect to solving the predictive task. Given multiple trained models to compare, we call the set of units classified as place or head direction (HD) cells $\mathcal{S}_m$ – classification method described in Section 3 – for each model $m$. Then, a subset $\mathcal{S}_m^*$ of units were lesioned at random from $\mathcal{S}_m$, keeping $|\mathcal{S}_m^*| = \min_m |\mathcal{S}_m|$. Lesioning was performed by setting to zero selected units' input-to-hidden weights $W_x$ and hidden-to-hidden weights $W_h$. Finally, validation loss was calculated. When $|\mathcal{S}_m| > |\mathcal{S}_m^*|$, the process was repeated 100 times.

**Spatial representational similarity analysis (sRSA) and Isomap**  sRSA was used to quantify correspondence between neural and spatial distances. Given hidden layer's activity $A_t \in \mathbb{R}^{N_h}$ and position $\mathbf{p}_t$ at time $t$ for validation sequences of length $T$ (720 in our dataset), distances were calculated as

$$
\begin{aligned}
D_A &= 1 - \frac{A_t \cdot A_{t'}}{|A_t|_2 \, |A_{t'}|_2} \\
D_p &= |\mathbf{p}_t - \mathbf{p}_{t'}|_2
\end{aligned}
\tag{12}
$$

at all pairs of timesteps $t$ and $t'$, hence $\{D_A, D_p\} \in \mathbb{R}^T$. Finally, sRSA was defined as the Spearman rank correlation coefficient between the two distances $\rho(D_A, D_p)$. Isomap was used to obtain a two-dimensional visualisation of neural manifolds. 7,500 timesteps were randomly sampled from the validation set, the cosine distance was used, and the number of neighbours was set to 100. Each point in the Isomap visualisations was coloured by the l1-norm of its corresponding position $|\mathbf{p}_t|_1$.

# E  SPATIAL TUNING ANALYSIS

Once trained, the RNN hidden units were analysed to gauge their spatial tuning. For each locomotion stage, 9 sequences (4 of which were also part of the validation set) of 720 frames were used to measure spatial representations. We call $R^{(k)}$ and $P^{(k)}$ the rate map and polar map of unit $k$, respectively. A rate map shows how active a unit is in different locations within the environment, and their calculation is introduced in Section 3. A polar map shows how active a unit is as a function of the agent's head direction. The head direction space is divided into $B_\theta = 60$ bins and $P^{(k)} \in \mathbb{R}^{B_\theta}$ was calculated as

$$
P_i^{(k)} = \frac{\sum_t A_t^{(k)} \mathbb{1}_{\theta_t \in \mathrm{Bin}_i}}{\sum_t \mathbb{1}_{\theta_t \in \mathrm{Bin}_i}}, \quad i \in \{1, \dots, B_\theta\}
\tag{13}
$$

Where $A_t^{(k)}$ is the hidden layer's activity (or value) of unit $k$ and $\theta_t$ is the agent's head direction at time $t$. Numerator and denominator were smoothed using a Gaussian filter with a standard deviation of 5 bins.

**Positional selectivity**  To measure the positional selectivity of a unit, we calculated the spatial information (Skaggs et al., 1996) of the rate map corresponding to unit $k$ as

$$
\begin{aligned}
\mathrm{SI}_r^{(k)} &= \sum_{i,j} \left[ \frac{O_{i,j}}{\|O\|_1} \frac{R_{i,j}^{(k)}}{\hat{R}^{(k)}} \log_2 \left( \frac{R_{i,j}^{(k)}}{\hat{R}^{(k)}} \right) \right] \\
\hat{R} &= \frac{\sum_{i,j} R_{i,j}^{(k)} O_{i,j}}{\|O\|_1} \\
O_{i,j} &= \sum_t \mathbb{1}_{\mathbf{p}_t \in \mathrm{Bin}_{(i,j)}} \\
i,j &\in \{1, \dots, B_p\} \times \{1, \dots, B_p\}
\end{aligned}
\tag{14}
$$

where $O \in \mathbb{R}^{B_p \times B_p}$ is the occupancy matrix, $\hat{R}^{(k)}$ is the average activity of the unit normalized by the agent's occupancy, and $\mathbf{p}_t = (x_t, y_t)$ is the position of the agent at time $t$.

**Directional selectivity** To quantify directional selectivity, the resultant vector length (RVL) of the polar map corresponding to unit $k$ was calculated as

$$\text{RVL}^{(k)} = \frac{|\mathbf{r}^{(k)}|}{\sum_i P_i^{(k)}}$$

$$\mathbf{r}^{(k)} = \sum_i \exp\left(j \frac{2\pi}{2B_\theta} i\right) P_i^{(k)} \tag{15}$$

$$i \in \{1, \ldots, B_\theta\}$$

where $\mathbf{r}^{(k)}$ is the resultant vector, and $j$ is the imaginary unit. Spatial information of $P^{(k)}$ was calculated as

$$\text{SI}_d^{(k)} = \sum_i \left[\frac{D_i}{|D|_1} \frac{P_i^{(k)}}{\hat{P}^{(k)}} \log_2\left(\frac{P_i^{(k)}}{\hat{P}^{(k)}}\right)\right], \quad \{1, \ldots, B_\theta\} \tag{16}$$

where $D \in \mathbb{R}^{B_\theta}$ is the directional occupancy vector, and $\hat{P}^{(k)}$ is the average activity of the unit normalized by the agent's directional occupancy.

**Spatial metrics min-max normalisation** We applied min-max normalisation to spatial metrics $\text{SI}_r$, RVL, and $\text{SI}_d$ for all model-experiment comparisons. This procedure followed the standard approach: subtracting the minimum and dividing by the range (difference between maximum and minimum values). Minimum and maximum values were determined across all developmental stages. For experimental data normalisation, minimum values could also be derived from the "crawl" stage of simulated data, since no hippocampal recordings were available from animals at this early loco-motor stage.

### E.1 MODEL-EXPERIMENTAL CORRESPONDENCE VALIDATION

We validated model-experiment correspondence by comparing Jensen-Shannon (JS) distances be-tween model and experimental distributions. For metric $m \in \{\text{SI}_r, \text{SI}_d, \text{RVL}\}$, we estimated prob-ability distributions of experimental data $\mathbf{p}$ and modelled data $\mathbf{q}$ at each locomotion stage $c$ by calculating a histogram with 50 bins, excluding outliers (defined as being outside 1.5 times the interquartile range). We calculated the JS between these distributions as:

$$\text{JS}^{(m)} = \sum_{c \in \{\text{walk,run,adult}\}} \text{JS}(\mathbf{p}_c^{(m)}, \mathbf{q}_c^{(m)}) \tag{17}$$

We then compared these distances to the distributions of distances obtained by shuffling metrics from modelled units 10,000 times.

Table A4: p-values for one-sided pairwise Wilcoxon rank-sum tests (Wilcoxon, 1992) with Benjamini-Hochberg correction (Hollander et al., 2013) testing for increasing differences of Fig-ure 4.

| Measure | Condition | $p$-value walk $\to$ run | $p$-value walk $\to$ adult | $p$-value walk $\to$ adult with grid cells |
|---|---|---|---|---|
| SIr | Model | 0.0055 | $6 \times 10^{-10}$ | $1.1 \times 10^{-5}$ |
| SIr | Experiment | $4.8 \times 10^{-16}$ | $2 \times 10^{-16}$ | – |
| SId | Model | 0.0013 | $9.3 \times 10^{-10}$ | $1.3 \times 10^{-7}$ |
| SId | Experiment | $9.6 \times 10^{-7}$ | $2 \times 10^{-16}$ | – |
| RVL | Model | 0.0016 | $2.5 \times 10^{-9}$ | $2.5 \times 10^{-7}$ |
| RVL | Experiment | $8.6 \times 10^{-10}$ | $2 \times 10^{-16}$ | – |
| Perc. place cells | Experiment | $7.5 \times 10^{-5}$ | 0.0016 | – |
| Perc. HD cells | Experiment | 0.037 | 0.450 | – |
| Perc. place + HD cells | Experiment | 0.017 | 0.337 | – |

# F    COMPARISON WITH ALTERNATIVE MODEL

## F.1    SPATIAL CELL DEVELOPMENT IN RATE OF CHANGE MODEL

The rate of change model was trained on crawl trajectories with extended inter-frame intervals (1-3 seconds) to control for predictive demand in the emergence of spatial representations. Figure A5 shows the rate of change model did not reach same levels of spatial tuning when compared to the original model – staying well below experimental values.

## F.2    LIKELIHOOD ANALYSIS

For metric $m \in \{\mathrm{SI}_r, \mathrm{SI}_d, \mathrm{RVL}\}$, we estimated probability distributions of experimental data $\mathbf{p}_c^{(m)}$ as explained in Section E.1 – with 100 bins, applying a softmax to deal with zero-probability values. Then, we calculated the log-likelihood of generating experimental data with a given model as:

$$\mathrm{LL}_{\mathrm{model}}^{(m)} = \sum_{c \in \{\mathrm{walk,run,adult}\}} \sum_{k \in [1,\ldots,K]} \ln\left(\mathbf{p}_c^{(m)} \cdot \mathbf{1}_k\right) \tag{18}$$

where $\mathbf{1}_k$ is a vector of zeros, with a one in the position corresponding to the bin in which the value of metric $m$ (for unit $k$) belongs to, and $K$ is the number of hidden units. To compare models, we computed the BIC as:

$$\mathrm{BIC}_{\mathrm{model}} = -2\,\mathrm{LL}_{\mathrm{model}}^{(m)} + d_{\mathrm{model}}\ln(K) \tag{19}$$

where $d$ is the degrees of freedom, which we set to 1 for the rate of change model (which corresponds to the parameter specifying the gap), and to 4 for the original model (which corresponds to the number of parameters for trajectory simulation).

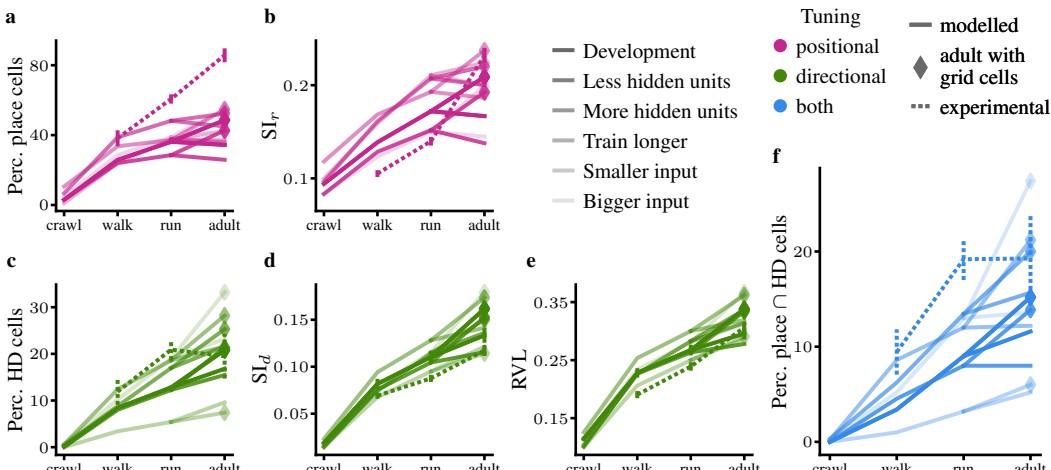

Figure A3: The maturation of spatial tuning does not deviate from the original results (indicated as "development") when decreasing/increasing hidden layer size, training longer, and feeding smaller/bigger images. Model data is shown in solid lines with transparency based on the changed parameter. Experimental data is shown with dotted lines. Diamonds indicate adult model trained with grid cell input. **a** Percentage of RNN units (solid lines) and percentage (mean±SEM) of hippocampal neurons (dotted line) classified as place cells. **b** Min-max normalized spatial information $SI_r$ (mean±SEM) for rate maps of RNN units (solid) and hippocampal neurons (dotted). **c** Percentage of RNN units (solid) and percentage (mean±SEM) of hippocampal neurons (dotted) classified as head direction (HD) cells. **d** Min-max normalised spatial information $SI_d$ (mean±SEM) for polar maps of RNN units (solid) and hippocampal neurons (dotted). **e** Min-max normalised resultant vector length (RVL, mean±SEM) for polar maps of RNN units (solid) and hippocampal neurons (dotted). **f** Percentage of RNN units (solid) and percentage (mean±SEM) of hippocampal neurons classified as both place and HD cells (dotted).

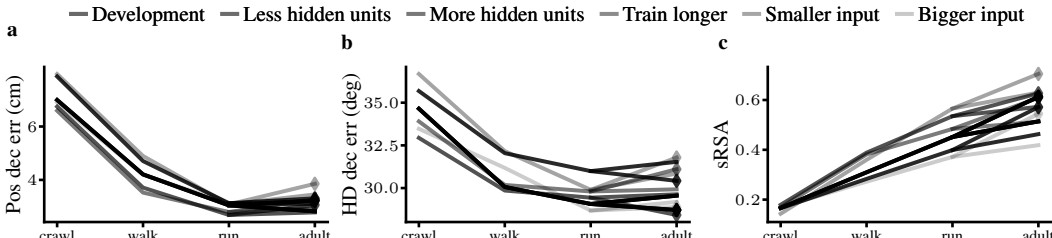

Figure A4: Decoding errors for position and head direction from hidden-layer activity decrease and correlation between agent's position and hidden state activity increases – as in the original model (development) – when performing a parameter sweep. Transparency indicates a change in parameters. Diamonds indicate adult model trained with grid cell input. **a** Position decoding errors (mean±SEM) and **b** head direction decoding errors (mean±SEM) calculated by linear regression on hidden units (50-50 train-test split on validation data). **c** Spatial representational similarity analysis (mean±SEM) quantifying correlation between agent position and hidden unit activity.

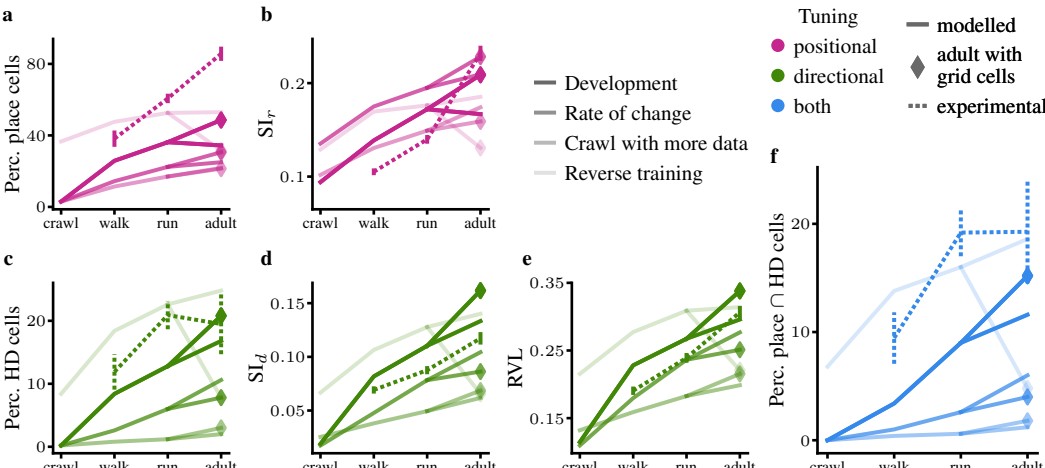

Figure A5: Spatial tuning development of alternative hypotheses controls deviates from original model results (development) and does not mimic experimental data (dotted lines). Model data is shown in solid lines with transparency indicating the control runs. Diamonds indicate adult model trained with grid cell input. **a** Percentage of RNN units (solid lines) and percentage (mean±SEM) of hippocampal neurons (dotted line) classified as place cells. **b** Min-max normalized spatial information $SI_r$ (mean±SEM) for rate maps of RNN units (solid) and hippocampal neurons (dotted). **c** Percentage of RNN units (solid) and percentage (mean±SEM) of hippocampal neurons (dotted) classified as head direction (HD) cells. **d** Min-max normalised spatial information $SI_d$ (mean±SEM) for polar maps of RNN units (solid) and hippocampal neurons (dotted). **e** Min-max normalised resultant vector length (RVL, mean±SEM) for polar maps of RNN units (solid) and hippocampal neurons (dotted). **f** Percentage of RNN units (solid) and percentage (mean±SEM) of hippocampal neurons classified as both place and HD cells (dotted).

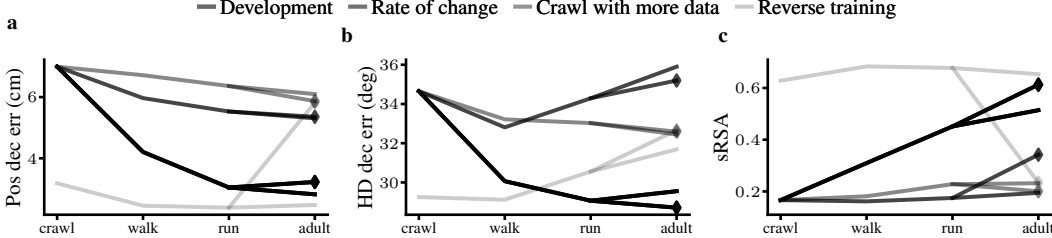

Figure A6: Decoding errors for position and head direction from hidden-layer activity and correlation between agent's position and hidden state activity remain stable in control runs. Original model results are indicated as "development". Transparency indicates control runs. Diamonds indicate adult model trained with grid cell input. **a** Position decoding errors (mean±SEM) and **b** head direction decoding errors (mean±SEM) calculated by linear regression on hidden units (50-50 train-test split on validation data). **c** Spatial representational similarity analysis (mean±SEM) quantifying correlation between agent position and hidden unit activity.

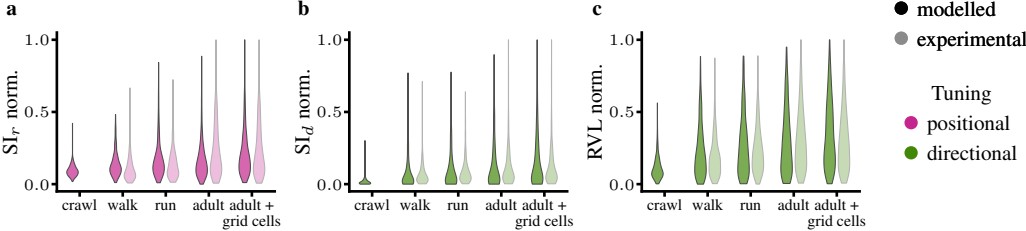

Figure A7: Population-level spatial metrics of modelled units and experimentally recorded ensembles of cells (dimmed colours), grouped by the rat's locomotion stage at the age of recording. **a** Min-max normalized spatial information $SI_r$ distribution for rate maps of RNN units (solid) and hippocampal neurons (dimmed). **b** Min-max normalised spatial information $SI_d$ distribution for polar maps of RNN units (solid) and hippocampal neurons (dimmed). **c** Min-max normalised resultant vector length (RVL) distribution for polar maps of RNN units (solid) and hippocampal neurons (dimmed).

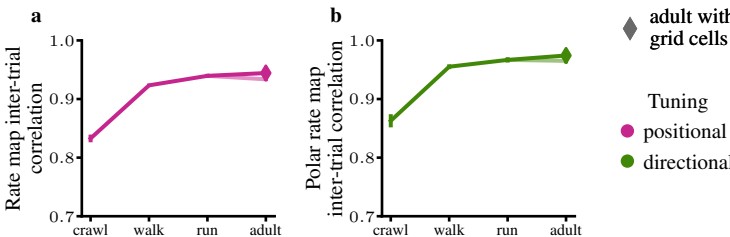

Figure A8: Inter-trial stability: Pearson correlation between **a** rate maps and **b** polar rate maps, computed on the first and second halves of the held-out validation trajectories across developmental stages. Diamonds indicate adult model trained with grid cell input.

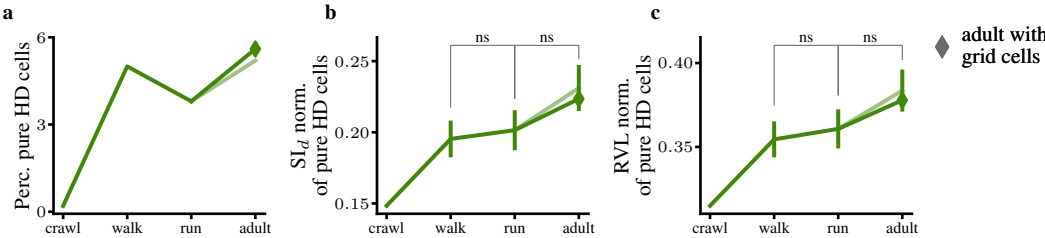

Figure A9: Pure directional selectivity emerges at the "walk" stage and remains stable thereafter. Diamonds indicate adult model trained with grid cell input. Non-significance was determined by one-sided pairwise Wilcoxon rank-sum tests (Wilcoxon, 1992) with Benjamini-Hochberg correction (Hollander et al., 2013) testing for increasing differences. **a** Percentage of RNN units classified as pure head direction (HD) cells – i.e., meeting only the criteria for directional selectivity. **b** Min-max normalised spatial information $SI_d$ (mean±SEM) for polar maps of RNN units classified as pure HD cells. **c** Min-max normalised resultant vector length (RVL, mean±SEM) for polar maps of RNN units classified as pure HD cells.

