# OpenReview forum: "From movement to cognitive maps: recurrent neural networks reveal how locomotor development shapes hippocampal spatial coding"
_ICLR.cc/2026/Conference — ICLR 2026 Oral_

### Official Review · Reviewer_Mf5d · 2025-10-23

**Soundness:** 4
**Presentation:** 4
**Contribution:** 4
**Rating:** 10
**Confidence:** 5

**Summary:**

This paper presents a compelling computational account of how locomotor
development shapes hippocampal spatial representations. The authors identify
distinct developmental locomotion stages through clustering analysis, then show
that RNNs trained sequentially on these movement patterns recapitulate the
biological timeline of spatial cell emergence. Notably, the model predicts
developmental increases in conjunctive place-direction cells, which the authors
confirm in experimental data.

**Strengths:**

* Rigorous methodology with excellent reproducibility
* Novel mechanistic insight linking embodied experience to neural development
* Strong experimental validation across multiple datasets
* Comprehensive controls ruling out alternative explanations
* Makes and validates testable predictions

**Weaknesses:**

## Minor Issues
* Missing ICLR requirement: The paper lacks the required statement about LLM usage in manuscript preparation. Please add this declaration.
* Grid cell specification: The grid cell input for the adult model (Eq. 2, Table 3) could be better justified. Please clarify: (a) what the scale values (0.2, 0.4, 0.6) represent in physical units, (b) how these relate to biological grid scales (given that they are usually increased from one scale to the next by a factor of something close to √{2}, and (c) rationale for the specific orientation value of 0.1 (although I guess this is due to the grid cell orientation relative to walls).
* Statistical reporting: Consider reporting exact p-values rather than just significance markers, particularly for borderline cases (e.g., Figure 4c,g "ns").

## Minor clarifications:
* Explain the min-max normalization approach more explicitly when first introduced
* In Figure 1f, "rat's eye view" should clarify this is simulated panoramic camera perspective (or if this is not a panoramic camera model, which camera model was used)

**Questions:**

1) Have you considered testing the model's predictions about conjunctive coding emergence through targeted experimental manipulations of locomotor development?
2) Could the framework extend to other sensory modalities or species with different developmental timelines?

---

> ### Author Response · Authors · 2025-11-27
> **Response part 1**
>
> We thank the reviewer for the enthusiastic endorsement and valuable feedback. We appreciate reading that rigorous methodology, novel insights, strong experimental validation, and comprehensive controls were listed as strengths of our manuscript. Moreover, we are grateful the reviewer did not report any major weakness. We will address the minor issues and respond to the questions below.
>
> _LLM statement_
>
> We appreciate the reviewer's attention and acknowledge that this requirement was inadvertently omitted from our submission. We have now prepared the declaration for inclusion in the revised manuscript: "Large language models (LLMs) were employed exclusively to improve the quality of written English and for stylistic refinement, with attentive supervision by the authors. No scientific content, experimental design, data analysis, or any intellectual contribution were generated through LLMs, nor did such assistance influence any reported results".
>
> ---
> _"(a) what the scale values (0.2, 0.4, 0.6) represent in physical units, (b) how these relate to biological grid scales [...] (c) rationale for the specific orientation value of 0.1 [...]"_
>
> We thank the reviewer for this question regarding the grid cell parameterization. As detailed in Section 3.1 and Appendix C, the scale values (0.2, 0.4, 0.6) represent the spacing between adjacent firing fields of grid cells. The physical unit is metres, within our simulated 0.625x0.625 m arena. These values fall within the biologically plausible range observed in the rodent medial entorhinal cortex (Stensola et al. 2012). While we acknowledge the canonical relationship between consecutive grid scales, our implementation employs a simplified scheme without deviating from biological plausibility, ensuring grid cells better tile our environment. Regarding the orientation parameter of 0.1 radians, this value specifies the orientation relative to walls, as the reviewer correctly notes. The specific value was selected to introduce an angular deviation from perfect alignment with the walls, as observed in biological grid cells.
>
> ---
> _"Report exact p-values, particularly for borderline cases (e.g. Figure 4c, g "ns")"_
>
> We thank the reviewer for the interest in the exact p-values, we will add the following table in the Appendix:
>
> | Measure  | Condition  | p-value walk → run | p-value walk → adult | p-value walk → adult with GC |
> | --- | --- | --- | --- | --- |
> | SIr | Model  | 0.0055 | $6 \times 10^{-10}$ | $1.1 \times 10^{-5}$ |
> | SIr  | Experiment | $4.8 \times 10^{-16}$ | $2 \times 10^{-16}$ | –  |
> | SId | Model | 0.0013  | $9.3 \times 10^{-10}$ | $1.3 \times 10^{-7}$ |
> | SId  | Experiment | $9.6 \times 10^{-7}$ | $2 \times 10^{-16}$ | –   |
> | RVL | Model | 0.0016 | $2.5 \times 10^{-9}$ | $2.5 \times 10^{-7}$ |
> | RVL | Experiment | $8.6 \times 10^{-10}$ | $2 \times 10^{-16}$ | – |
> | Percentage place cells | Experiment | $7.5 \times 10^{-5}$ | 0.0016  | – |
> | Percentage HD cells | Experiment | 0.037   | 0.450 | –   |
> | Percentage place + HD cells | Experiment | 0.017 | 0.337   | – |
>
> ---
> _"Explain min-max normalization approach when first introduced"_
>
> Min-max normalisation follows the standard approach of subtracting the minimum and dividing by the difference between the maximum and the minimum. For rate maps, min and max were calculated across all environmental bins, for spatial metrics normalisation was done taking min and max across the whole developmental pipeline. For spatial metrics of experimental data, the minimum could also belong to the simulated "crawl” developmental stage, because no animal with neural recordings belonged to that locomotor bin. We will include formulas for min-max normalisation of rate maps and spatial metrics in the Appendix and cite the relevant sections when first mentioning them in the main text.
>
> ---
> _"In Figure 1f, "rat's eye view" should clarify this is simulated [...]"_
>
> The reviewer is absolutely right, it is a panoramic camera view, we will include a note in our revision.

---

> > ### Author Response · Authors · 2025-11-27
> > **Response part 2**
> >
> > ---
> > _"Have you considered testing the model's predictions about conjunctive coding emergence through targeted experimental manipulations of locomotor development?"_
> >
> > We thank the reviewer for this suggestion and we acknowledge it represents a very valuable point. We would like to clarify that our work already contains control experiments that directly address this question. As detailed in Section 3.4 of our manuscript, we controlled for inter-frame temporal distance, cumulative training exposure, and developmental sequence ordering. These experiments demonstrate that the specific statistics of locomotor development—rather than confounding factors—drive the emergence of spatial representations, including the emergence of conjunctive cells. Moreover, as noted in the Discussion, our work directly generates testable predictions: since we demonstrated that locomotion development drives the emergence of hippocampal spatial tuning, we predict that early disruption of normal locomotion would impair spatial representation development in the hippocampus. This prediction is supported by one of our control experiments, where restricting the model to crawl-stage movements prevented the emergence of spatial representations.
> >
> > ---
> > _"Could the framework extend to other sensory modalities or species with different developmental timelines?"_
> >
> > We thank the reviewer for the very insightful question. Indeed, the framework we have developed can accommodate investigations across multiple sensory modalities and diverse species with varying developmental trajectories. Our current work represents an initial exploration focusing on developing rats, yet the underlying computational principles—namely, the transformation of egocentric sensory experiences into allocentric spatial representations through predictive learning—are general and allow broader application. Our ongoing research includes extensions to multimodal sensory integration, among other directions. We are currently investigating audio-visual integration using the ferret as an animal model of reference, selected for its auditory capabilities. While our ferret studies currently employ adult subjects, we envision a natural progression toward developmental research that would examine how maturation timelines might differentially shape the emergence of multimodal spatial representations.
> >
> > ---
> > **Finally**, we thank the reviewer again for the very positive assessment and we remain available to answer any additional questions.

---

### Official Review · Reviewer_HkDn · 2025-10-26

**Soundness:** 4
**Presentation:** 4
**Contribution:** 3
**Rating:** 8
**Confidence:** 4

**Summary:**

This paper examines how motor development of rodents may shape the spatial representations found in the brain. By analyzing real rodent behavior during development, the authors show that their trajectories can be broadly classified into 3 stages. The authors then train RNNs on predicting the next sensory and motor state, given the current sensory and motor state. They use sensory inputs that are inspired by real experiments, as well as real rodent trajectories. Analyzing the learned representations from the RNN, the authors find that place and head direction cells emerge with training, mirroring what is found in experiments. The authors also show that the model predicts a greater increase in direction selective place cells, which they find evidence for when re-analyzing existing neurophysiology data.

**Strengths:**

1. This paper is - to my knowledge - the first computational work to show how motor development impacts spatial representations. This is an important point, and one that will (and should) change how the field thinks about place and head direction cells.

2. The paper is well written and easy to follow.

3. The experiments are well done, clearly explained, and aligned with experimental evidence.

4. The use of the RNN model to make a prediction about spatial development that is then found in re-analyzed neurophysiological data is great and further strengthens the case for using RNNs to study this.

**Weaknesses:**

I identified no major weaknesses of this submission. However, there are a few things that I think should be addressed to make the paper stronger:

1. The authors train the RNN to do one time-step ahead prediction of both motor and sensory state. For path integration, this kind of one step prediction makes sense to me. E.g. when it's dark and I walk from my bed to my bathroom, I'm constantly updating where I think I am. But it was unclear to me: a) why this kind of one time-step prediction would be done in the hippocampus with sensory information b) why this prediction should be one time-step ahead (as opposed to other time-scales, or multiple time-scales). Adding more discussion on this to provide better motivation would be helpful I think.

2. The authors correctly say that their approach for identifying place and head direction cells - based on a spatial information metric - is standard. However, there is a growing understanding in the systems neuroscience field that a fixed threshold may not be the best way to classify. Many labs now use measures of cross-validation, robustness, consistency, etc., as well as compare to shuffled controls, to identify place cells. I understand the authors are primarily comparing to older experiments where this fixed thresholding approach was used, but it would be helpful I think to include some quantification of robustness of the place and head direction cells. If you split the trial in half and compute the SI for each half, is it similar?

3. Related to the point above, it would be helpful to have population summary level plots of the distribution of spatial information and RVL, in addition to having individual units plotted (Fig. 3). This would help the readers understand the extent to which the RNN model develops these kinds of properties.

4. Border cells seem to emerge in both the rodent recordings and the RNN model. Does the border score (and percent border cells) also increase in a similar way between RNN and neurophsyiological data? If so, this could be a further strengthening of the results (and could potentially be a supplemental figure). If no, then this suggests something different between the RNN and rodent development that would be interesting to comment on.

5. Finally, Cueva and Wei (2019) looked at how an RNN trained to perform path integration develops spatial representations. They find that HD units emerge first, then border units, then grid units. This is in rough agreement with actual experiments. While Cueva and Wei (2019) did not consider how properties of trajectories the RNN follows impacts the development (as the authors of this submission do), I do think it is reasonable for the authors to acknowledge this prior work and its attempt to study development of spatial representation through RNNs.

**Questions:**

1. Why do the authors choose the loss function they do (one time-step ahead prediction)?

2. How robust are the place and HD cells the model produces?

3. What are the population level properties of the RNN units (i.e., distribution of SI, distribution of RVL)?

4. How does border score emerge with training?

---

> ### Author Response · Authors · 2025-11-27
> **Response part 1**
>
> We thank the reviewer for the strong endorsement and appreciation of the importance of the scope of our work, clarity of presentation, experimental rigour, and comparison with experimental data. Moreover, we are glad to read that the reviewer found no major weaknesses in our manuscript and we answer your points hereafter.
>
> _"train the RNN to do one time-step ahead prediction [...] a) why this kind of one time-step prediction would be done in the hippocampus with sensory information b) why this prediction should be one time-step ahead (as opposed to other time-scales, or multiple time-scales). [...]"_
>
> The reviewer raises an important question regarding the biological motivation for single-step sensory prediction and the temporal scale of prediction. Regarding point (a), we would highlight the well-established predictive learning framework that characterizes hippocampal function. As also detailed in our response to Reviewer _PPg2_, who raised a similar concern, this framework positions the hippocampal formation as a system that compares incoming sensory stimuli with memory-derived predictions (Eichenbaum et al. 2004; Levy 1989) and is supported by observations that hippocampal neurons encode not just current location but trajectories of possible future positions (Johnson et al. 2007, Kay et al. 2020, Ujfalussy et al. 2022). The integration of egocentric sensory information—particularly visual input—within this predictive framework is fundamental to constructing stable allocentric representations (Tolman, 1948). Indeed, the extensive literature we cite in the introduction shows that predictive processing has been proven to spontaneously generate hippocampal-like spatial tuning  across multiple computational frameworks (Stachenfeld et al., 2017; Buzsaki & Tingley, 2018; Cueva & Wei, 2018; Whittington et al., 2020; Recanatesi et al., 2021; Uria et al., 2022; Gornet & Thomson, 2024; Levenstein et al., 2024).
>
> Regarding point (b), we acknowledge that while multi-step prediction has proven valuable in some models (Levenstein et al. 2024), our single-step approach was sufficient to demonstrate the key developmental principle we investigated. Importantly, multiple models (Recanatesi et al. 2021, Uria et al. 2022, Gornet et al. 2024) showed that single-step visual prediction can generate hippocampal-like representations, supporting our choice. Our focus on isolating the effects of movement statistics required keeping other variables constant—adding multi-step prediction would have introduced confounding factors that could obscure our main finding. Nevertheless, this opens a very interesting new question for future investigation connecting our findings to hippocampal replay development. The developmental trajectory of rest replay events (periods of inactivity while performing an experiment) described by Muessig et al. (2019) provides particularly relevant context: younger rats exhibit predominantly stationary replay covering little or no distance, with both the mean distance covered by these events and their mean speed both gradually and linearly increasing during development. This progression suggests that the capacity for predicting (or imagining) longer trajectories may have an influence on the maturation of spatial representations during development. An hypothesis that would be very interesting to test in future studies by, for example, removing sensory inputs and driving the network's units with noise.
>
> ---
> _"[...] Many labs now use measures of cross-validation, robustness, consistency, etc., as well as compare to shuffled controls, to identify place cells. [...] If you split the trial in half and compute the SI for each half, is it similar?"_
>
> We thank the reviewer for this important methodological consideration. We fully agree that quantifying the robustness of spatial representations is essential for further validating our findings. We will include an additional figure in the appendix reporting population-level Pearson correlation analyses between rate maps (and polar rate maps) computed on the first and second halves of the held-out validation trajectories across developmental stages. As a result, our analyses confirmed that correlations consistently remain above 0.8 throughout development, demonstrating the stability of the spatial responses our model develops. Moreover, we observed that inter-trial correlation increased throughout developmental stages, mirroring experimental recordings from developing rats (Muessig 2013).

---

> > ### Author Response · Authors · 2025-11-27
> > **Response part 2**
> >
> > _"[...] population summary level plots of the distribution of spatial information and RVL, in addition to having individual units plotted (Fig. 3). [...]"_
> >
> > We thank the reviewer for this suggestion. Figure 4 provides population-level summary statistics (mean $\pm$ SEM) for spatial information and resultant vector length across developmental stages, while Figure 3 illustrates qualitative examples of individual unit tuning. However, we agree that showing the full distribution of these metrics would be valuable. We will include histograms or violin plots showing the complete distributions of spatial information and RVL across developmental stages in the appendix of the revised manuscript.
> >
> > ---
> > _"[...] Does the border score (and percent border cells) also increase in a similar way between RNN and neurophsyiological data? [...]"_
> >
> > We appreciate the reviewer's insightful suggestion regarding border cells, which indeed represents a fascinating avenue of investigation. We acknowledge that incorporating a comprehensive analysis of border cells would be a very meaningful contribution to our manuscript. However, such an addition would require considerable expansion beyond the scope of our current submission. A thorough treatment would mean not just quantifying border scores across development, but also establishing comparisons with the available developmental data, and providing appropriate discussion of the computational and biological implications of any observed pattern. Furthermore, integrating this analysis into our main results would involve substantial restructuring of this section, since relegating such addition to the appendix might undervalue its potential significance. However, we will add a sentence on this important future direction to our Discussion section.
> >
> > ---
> > _"Cueva and Wei (2019) looked at how an RNN trained to perform path integration develops spatial representations. [...] While Cueva and Wei (2019) did not consider how properties of trajectories the RNN follows impacts the development (as the authors of this submission do), I do think it is reasonable for the authors to acknowledge this prior work"_
> >
> > We thank the reviewer for highlighting this relevant prior work. We believe the reviewer is referring to Cueva & Wei (2018) published at ICLR, which indeed includes a section examining the developmental timeline of spatial tuning properties, please let us know if this is not the case. While we cite this publication in our introduction as one of the works demonstrating the effectiveness of recurrent networks in generating spatial tuning, we acknowledge having missed this relevant section about the timeline of spatial response emergence. However, a critical distinction is that their analysis examined the emergence order across training epochs, in a less realistic task (path integration predicting x-y coordinates), using the same movement patterns, and without quantitative comparison with experimental data. Nonetheless, we will ensure that our expanded introduction more prominently acknowledges the order of spatial cell emergence observed in Cueva and Wei's paper.
> >
> > ---
> > **Finally**, we thank the reviewer for their thoughtful and constructive feedback. We greatly value their engagement with our work and suggested avenues for future expansions. We remain available to address any additional questions or concerns that may arise.

---

> > > ### Comment · Reviewer_HkDn · 2025-11-27
> > >
> > > I thank the authors for their detailed and clear response. The points addressed show that they considered the questions and I think the answers will help strengthen the paper.
> > >
> > > The point about development leading to greater capacity for predicting longer trajectories is very very interesting! It reminds me a little bit of the original work on curriculum learning (Elman 1993), where Elman showed that greater "attention" leads to better capacity to learning longer linguistic sequences.  I think this is an exciting future direction and one I hope the authors pursue.
> > >
> > > I do not have any additional questions. Best of luck on this paper!

---

### Official Review · Reviewer_Wj3x · 2025-10-30

**Soundness:** 2
**Presentation:** 2
**Contribution:** 1
**Rating:** 2
**Confidence:** 4

**Summary:**

This paper investigates the relationship between locomotor development and the emergence of spatial representations in the hippocampus. The authors employ a computational approach, analyzing rat locomotion data to define developmental stages (crawl, walk, run, adult) and training a recurrent neural network (RNN) model to predict visual input from previous movement states. The core hypothesis is that changes in locomotor development drive the sequential emergence of spatial tuning properties in the hippocampus.

**Strengths:**

The most notable strength of this work lies in its data-driven finding regarding the emergence of directional tuning. The analysis suggests that hippocampal directional tuning during development tends to arise from cells that initially encode location. This specific observation, derived from their data analysis, offers a potentially valuable insight into the developmental trajectory of hippocampal function.

**Weaknesses:**

My assessment of this paper reveals fundamental weaknesses that severely undermine its conclusions and overall scientific contribution.

1. Overly Strong and Unjustified Core Hypothesis: The central assumption that developmental changes in brain spatial representations are solely determined by locomotor patterns is a drastic oversimplification. This hypothesis completely neglects the crucial role of genetically programmed neural system development, which is an undeniable and fundamental aspect of brain maturation. To validate such a strong, almost certainly unreasonable assumption, the authors would need exceptionally compelling evidence, which is conspicuously absent.

2. Insufficient Evidence to Support the Core Hypothesis: The model, at best, can only capture some features of CA1 representational development. Crucially, the authors themselves acknowledge that their model fails to spontaneously generate grid cells, and struggles to adequately characterize the developmental features of head-direction (HD) cells in the medial entorhinal cortex (MEC) and other relevant brain regions. This demonstrates that their core hypothesis lacks the explanatory power to account for the full complexity of the navigation system's development. If locomotor patterns were the primary driver, the model should reproduce these key elements more robustly.

3. Biological Implausibility of Model Choice for CA1: The authors' argument that their RNN model is specifically modeling CA1 is highly problematic and fundamentally flawed.
(1) **Lack of Biological Constraints**: The RNN model is trained without any specific biological constraints pertinent to CA1. Without articulating why brain regions other than CA1 cannot be similarly modeled as RNNs, this claim is entirely unsubstantiated.
(2) **Contradiction with CA1 Connectivity**: CA1 is widely recognized for having very sparse recurrent connections; indeed, it is primarily considered a feedforward processing stage from CA3 and entorhinal cortex. Therefore, using a recurrent neural network (RNN), whose defining characteristic is its rich internal recurrence, to model a region known for its lack of such connections is biologically inconsistent and fundamentally unsound. This choice of model severely undermines any mechanistic insights the paper claims to offer regarding CA1 function.

**Questions:**

Could the authors provide insight into why their current framework might be insufficient for this emergence, especially when compared to other models, such as the Tolman-Eichenbaum Machine (TEM), which have successfully produced grid-like representations? Specifically, what fundamental architectural or mechanistic differences in TEM, if incorporated into the authors' model, might enable the emergence of grid cells?

---

> ### Author Response · Authors · 2025-11-27
> **Response part 1**
>
> We thank the reviewer for the thoughtful and detailed review. We are grateful to read that our data-driven findings were mentioned as the most notable strength of the paper. We address the reviewer’s questions and concerns below, suggesting some modifications to our manuscript that would hopefully clarify these points for future readers.
>
> _"Overly Strong and Unjustified Core Hypothesis [...] This hypothesis completely neglects the crucial role of genetically programmed neural system development [...]"_
>
> We thank the reviewer for this comment, which provides an opportunity to clarify our work's central argument. We respectfully note that at no point in our manuscript do we claim that locomotion is the "sole" determinant of spatial representations in the hippocampus. Our language is deliberately measured: we hypothesise that developmental experiences "fundamentally influence" (line 64), "shape" (line 98), and "play a key role in" (line 317) the emergence of spatial representations—explicitly acknowledging a contributory role rather than exclusive causation. We will revise the Introduction to make this framing clearer: while we acknowledge the fundamental importance of genetic programs and sensory development, our aim is to investigate a separate and previously unstudied contributor—specifically, the locomotor statistics with which the developing animal samples its sensory world.
>
> Regarding genetically programmed neural development, our framework is compatible with, and indeed assumes, such constraints. Hippocampal pyramidal neurons are generated between embryonic days E16–E21 (Bayer 1980), establishing the architecture upon which experience operates. Our model's structure—the RNN architecture, sensory input format, and learning objectives—can be understood as implementing inductive biases analogous to genetic programs. Recent work demonstrates that rearing environments shape spatial representations in the developing hippocampus (Farooq & Dragoi 2024; Ulsaker-Janke et al. 2023), and that birthdate-dependent preconfiguration interacts with experience-dependent refinement (Cavalieri et al. 2021; Huszár et al. 2022). Our contribution is to prove that within a constrained architecture, the specific temporal structure of locomotor experience accounts for key phenomena of spatial cell maturation.
>
> ---
>
> _"[...] The model, at best, can only capture some features of CA1 representational development. [...] their model fails to spontaneously generate grid cells, and struggles to adequately characterize the developmental features of head-direction (HD) cells in the medial entorhinal cortex (MEC) and other relevant brain regions."_
>
> We appreciate the reviewer's careful consideration of our model's scope. We would first note that prior to this work, no computational model proposed—let alone tested—whether locomotor development contributes to the emergence of spatial neurons. The fact that a deliberately simplified architecture, driven solely by realistic locomotor statistics, captures key features of hippocampal cell maturation is itself a novel finding. We do not claim that locomotion accounts for all developmental variability, but demonstrating that it accounts for a substantial portion, quantitatively matching experimental recordings, represents a significant advance in understanding spatial coding ontogeny.
>
> Regarding our model's scope: the RNN was deliberately simplified to isolate the influence of locomotor statistics while minimising confounds from architectural complexity. Our comparisons focused on CA1 because (i) this is the region for which developmental data were available, and (ii) as the anatomical output of the hippocampal formation, CA1 responses are indicative of the system's final processed output.
>
> Regarding head direction (HD) cells: to address this point, we performed additional analysis examining cells tuned only to HD (not position). Importantly, our model demonstrated adult-like HD tuning emerging at the "walk" stage, corresponding to approximately P16 (with instances at younger ages), and remained stable thereafter. This timeline is entirely consistent with experimental findings: mature HD signals appear around P15 in dorsal presubiculum and MEC (Wills et al. 2010; Langston et al. 2010), with immature HD signals observable as early as P12 (Tan et al. 2015; Bjerknes et al. 2015). The increase in HD-related metrics shown in our results (Figure 4c-e, g) predominantly reflects the emergence of conjunctive place-HD cells rather than pure HD cells—a distinction we will clarify in the revised Section 3.3, including an additional appendix figure showing the development of pure HD signal.

---

> > ### Author Response · Authors · 2025-11-27
> > **Response part 2**
> >
> > Regarding grid cells: their absence reflects our architectural simplification rather than a failure of the locomotor hypothesis. As discussed above, empirical evidence suggests grid cells depend on distinct mechanisms including attractor-like dynamics within MEC. We demonstrated that providing grid cell inputs to an adult variant of our model successfully reproduced adult-level spatial tuning, consistent with their late developmental emergence.
> >
> > ---
> > _"Biological Implausibility of Model Choice for CA1 [...] (1) Lack of Biological Constraints [...] (2) Contradiction with CA1 Connectivity [...]"_
> >
> > We appreciate the reviewer's attention to this point and acknowledge that our manuscript contained imprecise wording in a couple of locations (specifically in the first paragraph of Section 3.3 and line 580 of the Discussion), which may have contributed to this misunderstanding. We apologise for this confusion. To clarify: as stated in Section 3.1, our RNN serves as "an artificial hippocampus model"—it is not intended as a literal model of CA1 circuitry, but rather as a functional approximation of the hippocampal formation as a whole. Our experimental comparisons focused on CA1 recordings because this is the region for which we had access to developmental data; this does not imply that our model architecturally corresponds to CA1 alone. The revised manuscript will make this distinction explicit.
> >
> > (1) Regarding biological plausibility: we respectfully note that deep neural networks trained via backpropagation—a decidedly non-bioplausible learning rule—have nonetheless proved remarkably effective as functional models of brain systems, including the visual system (Yamins et al. 2014), motor cortex (Sussillo et al. 2015), and spatial memory (Cueva & Wei 2018; Whittington et al. 2020). Our work follows this established tradition: the goal is not to replicate biological implementation details, but to test whether a network solving a functionally appropriate task develops representations that match experimental observations. Specifically, our model implements predictive (self-supervised) learning, which characterises the hippocampal formation as a system that compares incoming sensory stimuli with memory-derived predictions (Eichenbaum et al. 2004; Levy 1989). This framework is supported by observations that hippocampal neurons encode not merely current location but trajectories of possible future positions (Johnson et al. 2007; Kay et al. 2020; Ujfalussy et al. 2022). The success of predictive RNNs in capturing hippocampal spatial coding across multiple studies, as documented in the introduction (Stachenfeld et al. 2017; Uria et al. 2022; Levenstein et al. 2024), validates this approach.
> >
> > (2) Regarding regional specificity: as noted above, the revised manuscript will clarify that our model captures hippocampal formation function more broadly, with experimental comparisons focused on CA1 due to data availability. This addresses the reviewer's valid concern about CA1's sparse recurrent connectivity.
> >
> > We look forward to including the reviewer’s feedback into our revision and we believe the manuscript will be clearer as a result.
> >
> > ---
> > _"why their current framework might be insufficient for this emergence [of grid cells], [...] compared to other models, such as the Tolman-Eichenbaum Machine (TEM)"_
> >
> > We appreciate the reviewer's interest in understanding the relationship between our model and the Tolman-Eichenbaum Machine (TEM; Whittington et al. 2020). The two models address fundamentally different questions. TEM was designed to capture relational and structural knowledge across discrete state spaces, using one-hot encoded abstract sensory cues and incorporating multiple architectural components: attractor networks, path-integration modules, and several loss functions (sensory prediction, memory retrieval, reconstruction, and regularisation). Our model, by contrast, was deliberately simplified to isolate a specific question: whether locomotor statistics alone can drive the emergence of spatial representations. We operate in continuous space with high-dimensional egocentric visual inputs precisely because this enables controlled manipulation of movement policies—the central variable of interest.

---

> > > ### Author Response · Authors · 2025-11-27
> > > **Response part 3**
> > >
> > > Had we incorporated TEM's architectural complexity, attributing developmental phenomena to locomotion rather than to specific features of the learning algorithm would have become considerably more difficult, if not impossible. The absence of grid cells in our model reflects this deliberate simplification rather than a failure of the locomotor hypothesis. As discussed in our response to Reviewer PPg2, empirical evidence suggests grid cells depend on distinct mechanisms—potentially attractor-like dynamics within MEC—that are not the focus of our developmental investigation.
> > >
> > > In summary, TEM and our model represent complementary rather than competing approaches: TEM addresses how abstract relational knowledge structures emerge, while our model addresses how naturalistic sensorimotor development drives spatial coding ontogeny.
> > >
> > > ---
> > >
> > > _Regarding contribution:_ We respectfully but firmly disagree with the assessment that our work lacks sufficient contribution. Prior to this work, no computational model had proposed—let alone tested—whether locomotor development contributes to the emergence of spatially tuned neurons. The developmental timeline of hippocampal spatial cells had been observed experimentally (lines 54–55) but remained mechanistically unexplained, with two competing hypotheses: intrinsic circuit maturation versus experience-dependent development (lines 62–63). Our work provides the first computational evidence supporting the latter, demonstrating that a deliberately simplified model driven by realistic locomotor statistics quantitatively matches key features of hippocampal spatial cell maturation.
> > >
> > > Crucially, our model generates novel predictions—specifically, the developmental increase in conjunctive place-direction coding—which we validate against hippocampal recordings. This represents the gold standard for computational neuroscience: propose a mechanism, derive predictions, and test them against experimental data. By constraining the hypothesis space and providing concrete avenues for future experimental validation (lines 29–31, 482–485), we believe this work makes a substantive contribution to understanding spatial coding ontogeny.
> > >
> > > ---
> > > **Given our responses above**, we respectfully ask whether the reviewer would consider revising the contribution score to better reflect the novelty and empirical validation of our findings—and possibly the overall rating of our submission.

---

### Official Review · Reviewer_PPg2 · 2025-11-01

**Soundness:** 4
**Presentation:** 4
**Contribution:** 1
**Rating:** 6
**Confidence:** 3

**Summary:**

This paper investigates how developmental changes in locomotion shape the emergence of spatial representations in the hippocampus. The authors analyzed experimental data on rat locomotor behavior, identified three developmental stages (crawl, walk, and run) via clustering. Then, the paper trained RNNs to predict visual input from visual and vestibular inputs using simulated trajectories that matched those observed in experimental data. They showed that the model exhibited the emergence of spatial tuned units in orders resembling the biological development timelines. The paper also provided a novel prediction where directional selectivity emerges through conjunctive place-direction coding and confirmed this in experimental data.

**Strengths:**

1. Originality: provided a novel mechanistic model connecting locomotion experience to hippocampal spatial neuron development, and provided novel predictions that are confirmed with experimental data.
2. Quality: well-controlled study with good ablation and control experiments (e.g., reversed developmental order).
3. Clarity: very clearly written. The figures are also informative and clear.
4. Significance: makes a substantive contribution to understanding how sensory-motor inputs shape spatial neuron formations in the hippocampus. It also opens an interesting direction for embodied AI, raising the question of how incorporating embodied inputs and developmental principles could improve the way AI systems learn useful internal representations for spatial tasks.

**Weaknesses:**

1. Biological motivation for the model architecture: The paper could provide a clearer biological rationale for the modeling choice. Why should we expect a single-layer RNN trained to reconstruct high-dimensional visual input to correspond to the hippocampal circuitry? Is there recurrent interactions between the spatially tuned neurons modeled in the paper? The author should better justify this correspondence.
2. Task: It would be useful to discuss why the paper chooses to predict the full high-dimensional visual input. Right now, it reads a bit arbitrary as an objective for studying spatially tuned neurons. Is the task choice critical for the results in the paper? Could an alternative task that is lower-dimensional or spatially-relevant (e.g., next location prediction?) yield similar results? Clarifying this could strengthen the argument that the finding reflects some general principle rather than task-specific results.
3. The context of prior computational work could be presented more clearly. The authors could provide a systematic comparison explaining what phenomena previous models have captured, what gaps remain, and how this work extends or differentiates itself.

**Questions:**

1. In the model, grid cell input is introduced as an external input provided only in the adult stage. Is it known that grid cells develop independently from the spatially tuned cells studied here? Could their development process be somewhat interactive?
2. How does the model and its learned representation generalize to novel maps or out-of-distribution visual conditions? Could the model predict developmental abnormalities if exposed to impoverished or atypical sensory inputs?

---

> ### Author Response · Authors · 2025-11-27
> **Response part 1**
>
> We thank the reviewer for the detailed and meaningful review. We appreciate the positive assessment of our work for its modelling originality, quality of experiments, clarity of presentation, and significance to both neuroscience and AI fields. We address the reviewer’s questions and concerns hereafter.
>
> _"Biological motivation for the model architecture [...]"_
>
> We thank the reviewer for this question, which allows us to clarify the biological grounding of our architectural choices. Our model is rooted in the predictive learning framework, which characterises the hippocampal formation as a system that compares incoming sensory stimuli with memory-derived predictions (Eichenbaum et al. 2004; Levy 1989). This framework is supported by observations that hippocampal neurons encode not just current location but trajectories of possible future positions (Johnson et al. 2007; Kay et al. 2020; Ujfalussy et al. 2022).
>
> Crucially, our modelling prioritises functional principles over anatomical detail. The single-layer RNN is not intended as a literal circuit model of any specific subregion; rather, it captures a fundamental computational principle—that recurrent dynamics enable the maintenance of internal states that predict future sensory input. The recurrence in our model abstracts over multiple sources of temporal integration in the biological system, including CA3 recurrent collaterals, the hippocampal-entorhinal loop, and broader cortico-hippocampal interactions. Multiple studies have demonstrated that predictive learning in such networks naturally produces spatially-tuned representations reflecting the statistical structure of experienced environments, as documented in the introduction (Stachenfeld et al. 2017; Cueva & Wei 2018; Whittington et al. 2020; Uria et al. 2022; Levenstein et al. 2024). This approach offers an alternative to pure attractor or path-integration models while incorporating elements of both (Recanatesi et al. 2021).
>
> We will revise the Introduction to expand on these biological motivations.
>
> ---
> _"[...] why the paper chooses to predict the full high-dimensional visual input. [...] Is the task choice critical for the results in the paper? Could an alternative task that is lower-dimensional or spatially-relevant (e.g., next location prediction?) yield similar results?"_
>
> We thank the reviewer for the question regarding task selection. To address the point about resolution-dependence, we highlight that our manuscript includes experiments where the network was trained on both lower (half) and higher (double) resolution visual frames as part of a parameter sweep. As shown in Figure A3, the developmental maturation of spatial tuning remained consistent across these conditions, confirming robustness to variations in input dimensionality.
>
> Regarding spatially-relevant tasks such as path-integration or next-location prediction: we deliberately avoided providing any direct spatial information (e.g. agent's position) to the model. This choice reflects the biological reality that egocentric sensory input is all the developing animal has direct access to—allocentric (i.e. world centred, such as place cells) spatial representations must be constructed from this “raw material”. Providing explicit allocentric coordinates as supervision would not only bypass the computational problem we sought to investigate, but could artificially bias the formation of spatial cell types rather than allowing them to emerge naturally from experience.
>
> ---
> _"The context of prior computational work could be presented more clearly. [...]"_
>
> We appreciate the reviewer's suggestion to more explicitly situate our work within the landscape of computational models of spatial learning. Various implementations have explored different aspects of this problem, including reinforcement learning frameworks (Stachenfeld et al. 2017), navigation through abstract spaces (Whittington et al. 2020), prediction from visual information (Uria et al. 2022; Gornet et al. 2024; Levenstein et al. 2024), integration of vestibular inputs (Cueva et al. 2018), and multimodal sensory combinations (Recanatesi et al. 2021).
>
> As rightfully noted by Reviewer HkDn, Cueva & Wei (2018) examined the emergence order of spatial cells across training epochs, observing head direction cells first, then border cells, and finally grid cells—in rough accordance with experimental data. However, several important distinctions apply. Their model was trained on path integration with explicit x-y coordinate prediction, providing the network with privileged access to allocentric spatial information that the developing animal does not possess. Crucially, their analysis used identical movement patterns during training, whereas the central question of our work is precisely whether and how the changing statistics of locomotor behaviour during development contribute to spatial cell emergence. Finally, their study did not include quantitative comparison with experimental recordings.

---

> ### Author Response · Authors · 2025-11-27
> **Response part 2**
>
> A critical gap therefore remains: no existing model has mechanistically linked the developmental progression of locomotor behaviour to the sequential emergence of spatially tuned neurons. While previous models have demonstrated that spatial representations can emerge from predictive learning in adult-like scenarios, they have not addressed how developmental changes in sensorimotor experience shape spatial coding ontogeny. We will ensure this distinction is clearly featured in the revised Introduction, alongside expanded discussion of our model's biological motivations.
>
> ---
> _"[...] Is it known that grid cells develop independently from the spatially tuned cells studied here? Could their development process be somewhat interactive?"_
>
> We thank the reviewer for raising this insightful question about the developmental relationship between grid cells and other spatial cell types. The empirical literature suggests that grid cells exhibit a distinct developmental trajectory: they are the last spatially-modulated neurons to emerge, with stable periodic firing first observable around P20 (Wills et al. 2010), and the transition to adult-like grid patterns occurs remarkably rapidly—over approximately 24 hours (Wills et al. 2012).
>
> Importantly, the evidence suggests an asymmetric dependence between place cells and grid cells. Brandon et al. (2011) demonstrated that medial septal inactivation, which abolishes grid cell firing, leaves hippocampal place fields largely intact. Conversely, Bonnevie et al. (2013) showed that hippocampal inactivation causes grid cell firing patterns to degrade. This indicates that place cells can function independently of grid cells, but not vice versa—supporting our approach of modelling place cell development without grid cell input during early stages.
>
> The rapid timescale of grid cell maturation, rather than the gradual weeks-long development seen for place cells, hints that grid cells may rely more heavily on intrinsic circuit mechanisms, possibly including attractor-like dynamics within MEC (Burak & Fiete 2009; Couey et al. 2013), rather than purely experience-dependent learning. While the precise mechanisms remain an active area of investigation, this distinct developmental profile supports our treatment of grid cell input as an external signal that comes online late in development. Future work could explore potential interactions during this transition period and in adulthood.
>
> ---
>
> _"[...] generalize to novel maps or out-of-distribution visual conditions? [...] if exposed to impoverished or atypical sensory inputs?"_
>
> This is a really interesting question. Our manuscript includes control experiments that address related concerns: we controlled for inter-frame temporal distance, cumulative training exposure, and developmental sequence ordering, demonstrating that the specific statistics of locomotor development—rather than confounding factors—drive the emergence of spatial representations.
> The broader question of environmental manipulation and novel sensory conditions represents a fascinating avenue for future investigation. Recent empirical work has documented how the rearing environment affects spatial representations in the developing hippocampus (Farooq & Dragoi 2024; Ulsaker-Janke et al. 2023), providing a rich set of experimental findings against which computational predictions could be tested. A comprehensive study examining the interaction between locomotor development and environmental manipulations—incorporating these documented outcomes—would constitute a substantial body of work in its own right, and one that would be better placed as future work. We will include discussion of this promising direction in the revised manuscript.
>
> ---
>
> **Finally, we note** that in the strengths section, the reviewer described our work as providing a "substantive contribution to understanding how sensory-motor input shapes spatial neuron development" and highlighted the "novel mechanistic model, predictions confirmed by experimental data." We are grateful for this positive assessment, but find it difficult to reconcile with a contribution score of 1 (poor).
>
> To summarise our novel contributions: this is the first computational work to mechanistically link developmental changes in locomotor behaviour to the sequential emergence of spatially tuned neurons—a phenomenon that has been observed experimentally but not previously explained. We validate these predictions quantitatively against hippocampal recordings, and our control experiments establish that movement statistics specifically, rather than confounds, drive this emergence.
> Given the strengths the reviewer has identified, we respectfully ask whether they would consider revising the contribution score to better reflect their assessment of the work's significance.

---

### Author Response · Authors · 2025-12-04
**Global Response**

**We would like to draw the Area Chair's attention to our general response.** Three of four reviewers recommended acceptance (ratings 10, 8, and 6), praising the "novel mechanistic insight", "rigorous methodology", and "strong experimental validation". **We note a critical inconsistency** from Reviewer PPg2 (who scored 6): despite rating our contribution as "1: poor", they explicitly described our work as providing a "substantive contribution to understanding how sensory-motor input shapes spatial neuron development" and highlighted our "novel mechanistic model, predictions confirmed by experimental data".

Only Reviewer Wj3x recommended rejection. **We believe their concerns (addressed below) arise from misreadings of our manuscript:**

- _Overly strong hypothesis that locomotion is the "sole" determinant of spatial representations in the hippocampus:_ At no point do we make this claim. Our language is deliberately measured: we hypothesise that developmental experiences "fundamentally influence", "shape", and "play a key role in" the emergence of spatial representations—explicitly acknowledging a contributory role rather than exclusive causation. While we recognise the importance of genetic programs, our aim is to investigate a separate and previously unstudied contributor: the locomotor statistics with which the developing animal samples its sensory world. We put forward a model, test it, and demonstrate that locomotion does contribute—quantitatively matching experimental recordings. Prior to this work, no computational model proposed—let alone tested—such an hypothesis, this represents a novel advance and not an "overly strong" claim.
- _Model scope, hippocampal formation vs CA1:_ As stated in Section 3.1, our RNN serves as "an artificial hippocampus model"—it is not intended as a literal model of CA1 circuitry, but rather as a functional approximation of the hippocampal formation as a whole. The recurrence abstracts over multiple sources of temporal integration in the biological system, including CA3 recurrent collaterals, the hippocampal-entorhinal loop, and broader cortico-hippocampal interactions. Our experimental comparisons focused on CA1 because this is the region for which developmental data was available and—as the anatomical output of the hippocampal formation—CA1 responses are indicative of the system's final processed output.

**Consensus strengths**

- First computational work to mechanistically link locomotor development to the emergence of spatially tuned neurons (PPg2, HkDn, Mf5d). As HkDn noted: "This is an important point, and one that will (and should) change how the field thinks about place and head direction cells".
- Novel predictions confirmed by hippocampal recordings (all reviewers).
- We received maximum scores for soundness and presentation from three reviewers, highlighting rigorous methodology, with clear and informative presentation.

**Key clarifications**

- _Biological motivation:_ Our model is rooted in the predictive learning framework, which characterises the hippocampal formation as a system that compares incoming sensory stimuli with memory-derived predictions (Eichenbaum et al. 2004; Levy 1989). This framework is supported by observations that hippocampal neurons encode not just current location but trajectories of possible future positions (Johnson et al. 2007; Kay et al. 2020; Ujfalussy et al. 2022). Multiple studies have demonstrated that predictive learning in recurrent networks naturally produces spatially-tuned representations, as documented in our Introduction.
- _Genetic programs:_ Hippocampal pyramidal neurons are generated between embryonic days E16–E21 (Bayer 1980), establishing the architecture upon which experience operates. Our model's structure—the RNN architecture, sensory input format, and learning objectives—can be understood as implementing inductive biases analogous to genetic programs.

**Revisions**

- Expanded biological motivation and contextualisation with prior work in Introduction.
- Clarification that model represents hippocampal formation as a whole, not CA1 specifically.
- New appendix figure showing pure HD signal development in the model, and consequent clarification of first paragraph of Section 3.3.
- Population-level violin plots of spatial metrics and inter-trial correlations proving robustness.
- Additional methodological details, p-value table, and LLM usage statement in Appendix.

**We respectfully request the Area Chair** to consider the following points when making their final recommendation: (i) the strong endorsement from the three reviewers that recommended acceptance, (ii) the internal inconsistency in Reviewer PPg2's evaluation—maximum scores for soundness and presentation, explicit praise for "substantive contribution", yet a contribution score of 1, and (iii) that Reviewer Wj3x's criticisms come from a misunderstanding of our central argument. We believe our comprehensive responses have addressed all concerns raised.

---

### Meta-Review · Area_Chair_WFg2 · 2025-12-18

**Summary:**

The reviewers were a bit polarized on the theoretical validity and biological plausibility of the model. While some praised the work for providing novel mechanistic insights connecting locomotion to hippocampal development, one reviewer strongly criticized the core hypothesis—that locomotion drives spatial representation—as an oversimplification that ignores genetic programming. This reviewer also argued that modeling the CA1 region as a Recurrent Neural Network (RNN) is fundamentally flawed because CA1 is biologically characterized by sparse recurrent connections and functions primarily as a feedforward stage. Conversely, others found the use of RNNs compelling, especially given the validation of the model's predictions against re-analyzed neurophysiological data.

On the technical side, questions were raised about the training objective and task design. Reviewers asked for a clearer justification of the one-step-ahead prediction task involving high-dimensional visual inputs, suggesting that lower-dimensional or more spatially relevant tasks might yield similar results. There were also calls for more rigorous quantification of the learned representations, including robustness checks (e.g., cross-validation of spatial information metrics) and population-level summaries of spatial tuning properties. Additionally, reviewers requested better explanations for specific modeling choices, such as the parameters for grid cell inputs and the rationale for using a single-layer RNN to model hippocampal circuitry.

**Reviewer Concerns:**

The authors provided a robust reponse to the critical reviewer. They pointed out that the reviewer was over-interpretting their claims about the role of locomotion in spatial representation formation, and provided good response regarding the concerns on genetic priors and biological realism of the model. Altogether, it is the AC's assessment that the authors addressed these concerns (though it is unclear whether the reviewer would agree).

For the other concerns, the authors did a good job of addressing the reviewer's comments.

**Reviewer Scores:**

The original scores were 6,2,8,10, and I suspect those would have at least been 7,3,8,10 after rebuttal.

---

### Decision · Program_Chairs · 2026-01-26

Accept (Oral)